# A Survey on the Possibilities & Impossibilities of AI-generated Text Detection

**Soumya Suvra Ghosal\***                                                    *sghosal@umd.edu*
*University of Maryland, College Park, MD, USA*

**Souradip Chakraborty\***                                                  *schakra3@umd.edu*
*University of Maryland, College Park, MD, USA*

**Jonas Geiping**                                                            *jgeiping@umd.edu*
*University of Maryland, College Park, MD, USA*

**Furong Huang**                                                             *furongh@umd.edu*
*University of Maryland, College Park, MD, USA*

**Dinesh Manocha**                                                           *dmanocha@umd.edu*
*University of Maryland, College Park, MD, USA*

**Amrit Singh Bedi**                                                         *amritbd@umd.edu*
*University of Maryland, College Park, MD, USA*

## Abstract

Large language models (LLMs) have revolutionized the domain of natural language processing (NLP) with remarkable capabilities for generating human-like text responses. However, despite these advancements, several works in the existing literature have raised serious concerns about the potential misuse of LLMs such as spreading misinformation, generating fake news, plagiarism in academia, and contaminating the web. To address these concerns, a consensus among the research community is to develop algorithmic solutions to detect AI-generated text. The basic idea is that whenever we can tell if the given text is either written by a human or an AI, we can utilize this information to address the above-mentioned concerns. To that end, a plethora of detection frameworks have been proposed, highlighting the *possibilities* of AI-generated text detection. But in parallel to the development of detection frameworks, researchers have also concentrated on designing strategies to elude detection, i.e., focusing on the *impossibilities* of AI-generated text detection. This is a crucial step in order to make sure the detection frameworks are robust enough and it is not too easy to fool a detector. Despite the huge interest and the flurry of research in this domain, the community currently lacks a comprehensive analysis of recent developments. In this survey, we aim to provide a concise categorization and overview of current work encompassing both the prospects and the limitations of AI-generated text detection. To enrich the collective knowledge, we engage in an exhaustive discussion on critical and challenging open questions related to ongoing research on AI-generated text detection.

# Contents

# 1 Introduction

In recent years, the natural language processing (NLP) community has witnessed a paradigm shift with the introduction of Large Language Models (LLMs) (Devlin et al., 2018; Brown et al., 2020; Chowdhery et al., 2022; OpenAI, 2022). Recently, ChatGPT (OpenAI, 2022), a chatbot based on the GPT-3 (Brown et al., 2020) model architecture, has attracted the attention of millions of users with its remarkable capability of generating coherent responses to a variety of queries. However, these advancements come as a double-edged sword to society. Specifically, the ease of using LLMs has worried the community about their potential misuse. For example, recent research has shown that LLMs can be used to generate fake news, spread misinformation, contaminate the web, or engage in academic dishonesty (Stokel-Walker, 2022). Additionally, LLMs could also be utilized for plagiarism, intellectual property theft, or the generation of fake product reviews, thereby misleading consumers and negatively impacting businesses (Chakraborty et al., 2023). Further, a recent study by Carlini et al. (2023) highlighted the risks and challenges related to the generation of synthetic training data using LLMs. Hence, although LLMs have led to a remarkable development in several tasks such as language translation (Vaswani et al., 2013; Huck et al., 2018; Radford et al., 2019; Brown et al., 2020), question-answering (Guu et al., 2020; Xiong et al., 2020), and text classification (Howard & Ruder, 2018) , mitigating their potential misuse is the need of the hour (Brown et al., 2020; Tamkin et al., 2021). Hence, a crucial question to address is: "*How can we harness the benefits of LLMs while also mitigating their potential drawbacks?*"

Even though it is difficult to answer the above question completely, a popular suggestion within the research community to address these ethical concerns is to accurately detect AI-generated text and separate it from human writings, thereby ensuring the responsible deployment of LLMs. Such a capability for AI detection would reduce the potential for misuse. Whether detection can effectively reduce harm is situation-specific. In some cases, even a limited capability to detect AI-generated text effectively combats misuse. For example, it can prevent accidental misattribution or make it more difficult to mislead consumers. In other situations, such as targeted fake news campaigns by motivated actors, only a strong capability to detect AI-generated text, would fully prevent misuse.

However, detecting AI-generated text is a challenging problem to solve in general. A study by Gehrmann et al. (2019) has shown that the ability of humans to detect AI-generated text was, unfortunately, only slightly better than a random classifier, even when tested against the language models available back in 2019. In light of this, researchers turned their attention to developing automated systems that can detect AI-generated text based on features that may not be easily recognizable by humans. Given the gravity of the problem, several works in the past few years have focused on the problem of AI-generated text detection. Common approaches include leveraging watermarking techniques (Aaronson, 2023; Kirchenbauer et al., 2023a; Zhao et al., 2023a; Kuditipudi et al., 2023b), designing detectors based on statistical metrics (Gehrmann et al., 2019; Mitchell et al., 2023) or fine-tuning classifiers on a set of training samples (Solaiman et al., 2019; Chen et al., 2023; Wu et al., 2023a). In contrast to the development of detection frameworks, interestingly, recent studies (Sadasivan et al., 2023; Krishna et al., 2023) have shown that state-of-art detection frameworks can be vulnerable and fragile to paraphrasing-based attacks. Further, Sadasivan et al. (2023) have also provided interesting theoretical insights about the impossibility of AI-generated text detection, outlining the fundamental impossibility of detecting a theoretically optimal language model. This analysis is followed by work elucidating the possibilities of AI-generated text detection in Chakraborty et al. (2023), which highlights that it should always be possible to detect AI-generated text via increasing the text length unless the distributions of human and machine-generated texts are the same over the entire support, i.e. the language model is theoretically optimal. This possibility result is also supported in subsequent work in Kirchenbauer et al. (2023b), showing that for the example of watermarking, empirically, attacks such as paraphrasing only dilute the signal that the text is AI-generated and that observing a sufficiently large amount of text makes detection possible again. However, how much text is available from a single source is application-specific. In this review, we provide a taxonomy of recent studies of this nature, discuss their interplay, detail their approaches to detection, outline their inherent limits, and identify the applications where specific techniques show promise.

### 1.1 Our Focus

Based on the recent research focus to address the problem of AI-generated text detection (AI-GTD), in this survey, we aim to provide a quick description of such recent results. We have divided the existing literature into two parts; *towards the possibility* and *towards the impossibility* of AI-GTD. The two categories are detailed as follows.

- **Towards the Possibilities of AI-GTD** (cf. Sec. 4): Under this category, we specifically review notable, recent frameworks designed to detect AI-generated text. These frameworks are categorized into two major groups based on usability: *prepared* (Sec. 4.1) and *post-hoc* (Sec. 4.2). We further classify *prepared* detectors into two main sub-groups based on the working principle: Watermarking-based (Sec. 4.1.1) and Retrieval-based (Sec. 4.1.4). Based on the sampling principle, watermarking detectors can be further granularized into biased-sampler watermarks and pseudo-random watermarks. On the other side, *post-hoc* detectors primarily encompass Zero-shot detection (Sec. 4.2.1) and detection based on fine-tuning classifiers (Sec. 4.2.2).

- **Towards the Impossibilities of AI-GTD** (cf. Sec. 5): Under this category, we review different attack schemes designed to evade detection. For the ease of readers, we subdivide the attacks into four major categories: Paraphrasing-based attacks 5.1, Copy-paste attacks 5.3, Generative attacks 5.2 and Spoofing attacks 5.4.

Figure 1 provides a pictorial representation of the above-mentioned categorization. In this survey, we aim to provide a detailed description of each of the results and also discuss some interesting open questions at the end. We start by providing a brief description of the functionality of language models and AI-GTD in Sec. 2. Then we proceed by reviewing the recent theoretical insights regarding AI-GTD explored by Chakraborty et al. (2023); Sadasivan et al. (2023) in Sec. 3. Next, we provide a comprehensive investigation of different studies highlighting the avenues and limitations of AI-GTD in Sec. 4 and Sec. 5, respectively. Finally, in Sec. 6, we provide a detailed systematic discussion on various critical questions related to the regulation of language models, designing robust test statistics, and accurate characterization of human distribution. Further, based on our discussion, we also emphasize future works to focus on designing fair and robust detectors immune to paraphrasing attacks.

## 2 Preliminaries: AI-generated Text Detection

In this section, we define the mathematical notation for the language model, and try to define the problem of AI-GTD concisely as follows.

**Language Models (LM).** Let us first define a language model mathematically. First, let us denote a vocabulary set as $\mathcal{V}$ and we denote a language model by a mapping $\mathcal{L}_\theta$ (parameterized by $\theta$). A language model $\mathcal{L}_\theta$ takes a sequence of tokens (called prompt) as input denoted by $\mathbf{h} := \{h_1, h_2, \cdots, h_N\}$, where each token $h_i \in \mathcal{V}$. The prompt is fed as input to the language model and it generates or predicts the first output token $s_0 \in \mathcal{V}$. To determine the next token $s_1$, the input prompt $\mathbf{h}$ and generated token $s_0$ are again fed as input to the language model as a new prompt $[\mathbf{h}, s_0]$. The process is reiterated for all time steps. Let the input prompt for the $t$-token be given by $[\mathbf{h}, \mathbf{s}_{1:t-1}]$ where $\mathbf{s}_{1:t-1} = \{s_0, s_1, \cdots, s_{t-1}\}$. In order to generate the next token for the given prompt, the LM outputs a $|\mathcal{V}|$-dimensional logit vector $\boldsymbol{\ell}_t$ corresponding to every token in vocabulary $\mathcal{V}$ such that $\boldsymbol{\ell}_t = \mathcal{L}_\theta([\mathbf{h}, \mathbf{s}_{1:t-1}])$. Then, a softmax function is used to map the logits $\boldsymbol{\ell}_t$ into a probability distribution over the vocabulary set $\mathcal{V}$ which we denote by $\mathbb{P}_{\mathcal{L}_\theta}(s_t = \cdot | [\mathbf{h}, \mathbf{s}_{1:t-1}])$. The probability of sampling token $s_i \in \mathcal{V}$ is given by $p_t(i) = \frac{\exp(\boldsymbol{\ell}_t(i))}{\sum_{v \in \mathcal{V}} \exp(\boldsymbol{\ell}_t(v))}$. Finally, the next token $s_t$ is generated by sampling from the distribution $p_t$. Some commonly used sampling techniques in the literature include Greedy Sampling, Random Sampling, Beam Search (Freitag & Al-Onaizan, 2017), Top-p sampling (Holtzman et al., 2019) and Top-k Sampling (Fan et al., 2018).

**AI-generated Text Detection.** In NLP, text detection problem has been commonly treated as a binary classification task (Jawahar et al., 2020). The main objective is to classify whether a candidate passage is

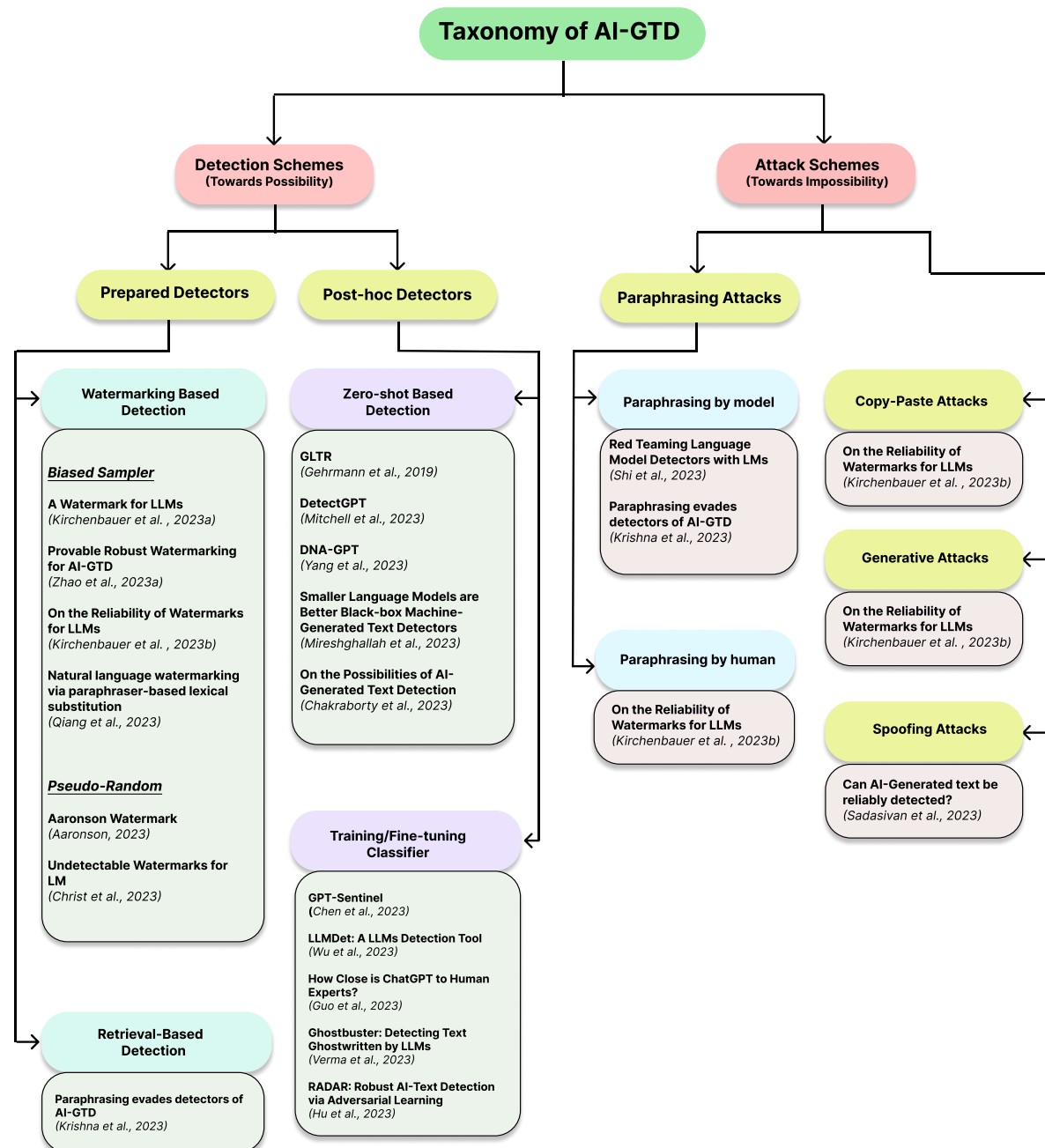

Figure 1: **Categorization of all explained methods.** In this survey, we have divided the existing literature into parts: *towards the possibility* and *towards the impossibility* of AI-GTD. As explained in Section 1, based on usability all detector frameworks can be categorized into two major groups: Prepared (Sec. 4.1) and Post-hoc (Sec. 4.2). Prepared frameworks can be further sub-categorized into four major groups based on the working principle: Watermarking-based detection (Sec. 4.1.1) and Retrieval-based detection (Sec. 4.1.4). Based on the sampling principle, watermarking detectors can be further sub-granularized into Biased Sampler watermarks and Pseudo-random watermarks. On the other branch, Post-hoc detectors primarily encompass Zero-shot detection (Sec. 4.2.1), detection based on fine-tuning classifiers (Sec. 4.2.2). For impossibilities, we review different attack schemes designed to evade detection.We subdivide the attacks into three major categories: Paraphrasing-based attacks 5.1, Copy-paste attacks 5.3, Generative attacks 5.2 and Spoofing attacks 5.4.

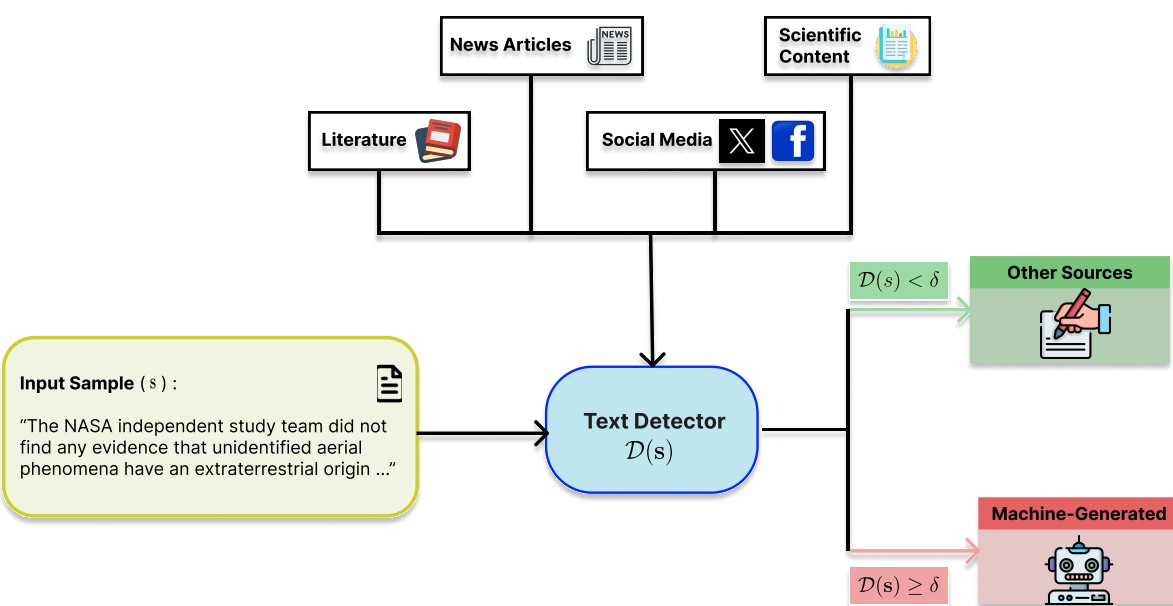

Figure 2: **Text Detection Framework.** This figure shows a flowchart adapted from Mitchell et al. (2023) to depict the general framework of AI-generated text detection.

AI-generated or human-written. Let us represent the set of all possible sequences as $\mathcal{S}$. Given an input sequence $\mathbf{s} \in \mathcal{S}$, a text detector $\mathcal{D}(\mathbf{s}) : \mathcal{S} \to \mathbb{R}$ outputs a scalar score, which is then compared against a threshold $\delta$ to perform detection. A score higher than the threshold $\mathcal{D}(s) \geq \delta$ serves as an indicator of AI-generated content, and conversely. A conceptual representation of the overarching architecture governing the identification of machine-generated text is portrayed in Figure 2.

## 3  Theoretical Insights of AI-Generated Text Detection

**Impossibility Result.** In a recent study, Sadasivan et al. (2023) argues that the current state-of-art AI-generated text detectors based on watermarking schemes (Kirchenbauer et al., 2023a) and zero-shot classifiers (Mitchell et al., 2023) are not reliable in practical scenarios. To support their argument, they show that a simple lightweight neural-network-based paraphraser such as PEGASUS (Zhang et al., 2020) can significantly reduce the detection accuracy of watermarking-based text detectors (Kirchenbauer et al., 2023a) without causing a drastic change in the perplexity score. Further, authors observed a 71% degradation in accuracy for zero-shot classifiers such as DetectGPT (Mitchell et al., 2023) when using a T5-based paraphraser. These observations highlight that even state-of-art text detectors are not immune to paraphrasing attacks, raising questions about their reliability.

To provide a theoretical understanding, Sadasivan et al. (2023) derived an impossibility result of AI-text detection. It states that when there is a strong overlap between the text distributions generated by humans and the model, even the best text detector performs marginally better than a random classifier. Formally, let $\mathcal{M}$ and $\mathcal{H}$ represent the text distributions of a model and human. The area under the ROC (AUROC) curve of a post-hoc detector (cf. Sec. 4.2) $\mathcal{D}$ can be bounded as:

$$\text{AUROC}(\mathcal{D}) \leq \frac{1}{2} + TV(\mathcal{M}, \mathcal{H}) - \frac{TV(\mathcal{M}, \mathcal{H})^2}{2}, \tag{1}$$

where $TV$ indicates the total variation distance between the two distributions defined as: $TV(\mathcal{H}, \mathcal{M}) = \max_E |\mathbb{P}_{s\sim\mathcal{H}}[s \in E] - \mathbb{P}_{s\sim\mathcal{M}}[s \in E]|$, for any event $E$. Now from the bound, when there is a strong overlap between machine ($\mathcal{M}$) and human ($\mathcal{H}$) generated text distributions, $TV(\mathcal{M}, \mathcal{H}) \to 0$. In this case, the upper bound on the AUROC of the detector approaches 1/2, representing a random classifier.

Watermarking solves this problem, as it can synthetically introduce a distribution shift between the two distributions. However, detection can still be made difficult by paraphrasing the text (Sadasivan et al., 2023; Krishna et al., 2023). Let $\mathcal{M}^*$ represent the distribution of the paraphrased samples. Then, for any watermarking scheme $W$, Sadasivan et al. (2023) provide the following bound:

$$\mathbb{P}_{\mathbf{s}\sim\mathcal{M}^*}[\mathbf{s} \text{ watermarked using W}] \leq \mathbb{P}_{\mathbf{s}\sim\mathcal{H}}[\mathbf{s} \text{ watermarked using W}] + TV(\mathcal{M}^*, \mathcal{H}) \tag{2}$$

If the paraphrased distribution $\mathcal{M}^*$ closely resembles the human text distribution $\mathcal{H}$, then $TV(\mathcal{M}^*, \mathcal{H})$ will be minimal. From the bound in Equation 2, this implies: (1) $\mathbb{P}_{\mathbf{s}\sim\mathcal{H}}[\mathbf{s} \text{ watermarked using W}]$ is high indicating an increase in false positives, i.e., misclassifying human-written text as AI-generated, or (2) $\mathbb{P}_{\mathbf{s}\sim\mathcal{M}^*}[\mathbf{s} \text{ watermarked using W}]$ is low indicating increase in misclassification of watermarked text as human-authored.

**Hidden Possibilities of Text Detection.** Interestingly, Chakraborty et al. (2023) argue that the impossibility results posited by Sadasivan et al. (2023) could be conservative in practical scenarios. Specifically, the impossibility result does not consider the potential number of text sequences available (the possibility of collecting multiple sequences) or the length of the text sequence, which can impact the detectability significantly, leading to a pessimistic conclusion. Additionally, the assumption in Sadasivan et al. (2023) that all human writings follow a "monolithic" language distribution is not realizable in practice. For example, Chakraborty et al. (2023) provides a motivating example in the critical case of detecting whether a Twitter account is AI-bot or human, and it's natural that in those scenarios, one can expect multiple samples to improve the detection performance. Chakraborty et al. (2023) derive a precise sample-complexity bound for detecting AI-generated text for both IID and non-IID settings, which highlights the hidden possibility in AI-generated text detection.

**IID Setting:** For the i.i.d scenario, given a collection of $n$ independent and identically distributed ($i.i.d$) text examples, $\{s_i\}_{i=1}^n$, they show that the AUROC can be bounded as:

$$\text{AUROC}(\mathcal{D}) \leq \frac{1}{2} + TV(\mathcal{M}^{\otimes n}, \mathcal{H}^{\otimes n}) - \frac{TV(\mathcal{M}^{\otimes n}, \mathcal{H}^{\otimes n})^2}{2} \tag{3}$$

where, $\mathcal{M}^{\otimes n}$ and $\mathcal{H}^{\otimes n}$ denotes the product distribution of model and human respectively. According to large deviation theory, the total variation distance ($TV(\mathcal{M}^{\otimes n}, \mathcal{H}^{\otimes n})$ approaches 1 at a rate exponential to the number of samples: $TV(\mathcal{M}^{\otimes n}, \mathcal{H}^{\otimes n}) = 1 - \exp(nI_c(\mathcal{M}, \mathcal{H}) + o(n))$, where $I_c(\mathcal{M}, \mathcal{H})$ represents the Chernoff information. Now, with an increase in the number of samples: $n \to \infty$, the total variation distance $TV(\mathcal{M}^{\otimes n}, \mathcal{H}^{\otimes n})$ approaches 1 exponentially fast thereby increasing the upper bound on AUROC. Thus, increasing the number of samples significantly improves the possibilities of AI-generated text detection. The sample complexity bound in Chakraborty et al. (2023) states that when the model and human distribution are $\gamma$ close to each other ($TV(\mathcal{M}, \mathcal{H}) = \gamma$), to achieve an AUROC of $\epsilon$, the number of samples required are:

$$n = \Omega\left(\frac{1}{\gamma^2}\log\left(\frac{1}{1-\epsilon}\right)\right). \tag{4}$$

The bound in Equation 4 indicates that the number of samples required to perform detection increases exponentially with an increase in the overlap between machine and human distribution. This observation has been further empirically highlighted in Chakraborty et al. (2023) (Figure 1).

**Non-IID Setting:** Similarly, Chakraborty et al. (2023) also extended the results to non-iid scenarios by showing that if human and machine distributions are close $\mathtt{TV}(m, h) = \gamma > 0$, then to achieve an $\mathtt{AUROC}$ of $\epsilon$, it requires

$$n = \Omega\left(\frac{1}{\delta^2}\log\left(\frac{1}{1-\epsilon}\right) + \frac{1}{\gamma}\sum_{j=1}^{L}(c_j - 1)\rho_j + \sqrt{\frac{1}{\gamma^2}\log\left(\frac{1}{1-\epsilon}\right) \cdot \frac{1}{\gamma}\left(\sum_{j=1}^{L}(c_j - 1)\rho_j\right)}\right) \tag{5}$$

number of samples for the best possible detector, for any $\epsilon \in [0.5, 1)$. where, as before $n$ represents the number of (dependent in this case) samples/sequence and $L$ represents the number of independent subsets where each subset is represented by $\tau_j$ with $c_j$ samples $\forall j \in (1, 2 \cdots, L)$. The non-iid result in (Chakraborty

et al., 2023) is more general and can be analyzed under different settings by varying the number of subsets with varying samples in each subset. The characterization of sample complexity results with Chernoff information as derived in Chakraborty et al. (2023) is novel and brings several new insights on how to design better detectors and watermarks.

To summarize, analysis in Sadasivan et al. (2023) highlights that when the model and human distribution overlap or are close to each other, i.e., when the total variation distance between the distributions is low, detection performance can only be random. However, when the number or length of samples available at detection time is taken into consideration, the analysis in Chakraborty et al. (2023) finds that for any level of closeness, a number of samples exist that provide strong detection performance. The theoretical claims were validated in (Chakraborty et al., 2023) for real datasets Xsum, Squad, IMDb, and Fake News dataset with state-of-the-art generators and detectors. The claims have been further validated also in the context of reliable watermarking in Kirchenbauer et al. (2023b). However, Sadasivan et al. (2023) highlights certain assumptions in these prior works that might not be practical for the detection scenarios. For example, improved detection performance with multiple iid samples as shown in the first part of work by Chakraborty et al. (2023) might not be practical in several scenarios like evaluation in exams, etc. Similarly, the defense by Krishna et al. (2023) requires storing the outputs of the LLM in a database, which, although efficient, might lead to privacy concerns. Additionally, with the possibility of spoofing and generative attacks in several real-world scenarios, it will be even harder to perform detection due to reduced TV distance.

Hence, we note that the claim of possibility or impossibility requires additional context about the problem, which is extremely important. Thus, a detailed description of various problem instances and detectability is presented in Sec. 6.6, where we discuss the detectability across different practical problem instances.

# 4 Towards the Possibilities of AI-generated Text Detection

In the light of ownership and practical usability, AI-generated text detectors can be categorized into two distinct groups: "Prepared" and "Post-hoc" detectors. Prepared detection scheme primarily include watermarking and retrieval-based detectors, which involves proactive involvement of the model proprietor during the text generation process. In contrast, Post-hoc detectors encompass zero-shot detection techniques or fine-tuned classifiers which can be used by external parties.

## 4.1 Prepared Detectors

### 4.1.1 Overview of Methods based on Watermarking

Although watermarking has been a long-known concept in the literature for hiding information within data, implementation was mostly challenging in the early days due to its discrete nature (Katzenbeisser & Petit-colas, 2016). Synonym substitution (Topkara et al., 2006), synthetic structure restructuring (Atallah et al., 2001), and paraphrasing (Atallah et al., 2001; 2002) were the commonly used approaches in the past for embedding a watermark into existing text (Zhao et al., 2023a). However, with advancement in neural language models (Vaswani et al., 2017; Devlin et al., 2018), rule-based approaches have been replaced with improved techniques based on using mask-infilling models (Ueoka et al., 2021). Recent approaches to watermarking include learning end-to-end models (Ziegler et al., 2019; Dai & Cai, 2019; He et al., 2022a;b) with both encoding and decoding of each sample. Watermarks specifically designed for large language models have recently gained major attention as a technique to detect machine-generated text (Aaronson, 2023; Kirchenbauer et al., 2023a; Zhao et al., 2023a; Christ et al., 2023a; Zhao et al., 2023b; Kirchenbauer et al., 2023b; Liu et al., 2023; Kuditipudi et al., 2023a; Zhang et al., 2023).

Formally, a general watermarking scheme can be defined as consisting of two probabilistic polynomial-time algorithms: `Watermark` and `Detect` (Zhao et al., 2023a). The `Watermark` algorithm takes in a language model $\mathcal{L}$ as input and modifies the model's outputs to encode a signal into generated text.

During inference, given a text sequence **s** and detection key $k$, the `Detect` algorithm outputs 1 if $s$ was generated by $\hat{\mathcal{L}}$ or 0 if it is generated by any other model. Ideally, a watermarking-based detector should exhibit the following properties: (1) Preserve the original text distribution, (2) Be detectable without access

to the language model, and (3) Be robust under perturbations and distribution shifts (Kuditipudi et al., 2023a). Christ et al. (2023a) also highlights that (4) the watermark should be ideally undetectable for any party not in control of the secret key that defines the watermark to make the detection widely and easily applicable.

Based on the sampling principle, watermarking schemes can be further sub-classified as either "biased-samplers", where the watermarked model's $\mathcal{L}$ output distribution is a modified probability distribution $\hat{p}_t = \mathbb{P}_{\hat{\mathcal{L}}_\theta}(s_t = \cdot|[\mathbf{h}, \mathbf{s}_{1:t-1}])$ on the vocabulary set $\mathcal{V}$, or as "pseudo-random" watermarks, where the model's output probability is formally not modified, but instead collapsed onto a particular sample from $\mathbb{P}_{\mathcal{L}_\theta}$, chosen pseudo-randomly based on local context.

We describe a number of watermarks from the first category in subsection 4.1.2, and of the second category in subsection 4.1.3.

### 4.1.2 Biased-Sampler Watermarks.

Biased-Sampler watermarking (Kirchenbauer et al., 2023a;b; Zhao et al., 2023a) modifies the token distribution at each time step to encourage the sampling of tokens from a pre-determined category (a `green` list at each token). As an advantage, since biased sampling creates a generic shift in the probability distribution, it can be coupled with any sampling scheme. However, biasing the output distribution can also lead to increased perplexity and degradation in text quality. Below, we summarize a few notable studies leveraging a biased-sample watermarking scheme to detect AI-generated text.

**A Watermark for Large Language Models (Kirchenbauer et al., 2023a)**  Kirchenbauer et al. (2023a) was the first to propose a watermark based on biased sampling from large language models and derive a matching detector for AI-generated text based on statistical metrics. Note that, watermarking language models require access to the generated probability distribution over the vocabulary at each time step. Let $\mathbf{s}_{1:t-1} = \{s_1, s_2, \cdots, s_{t-1}\}$ be the output of the language model till t-1-th step and $\ell_t \in \mathbb{R}^{|\mathcal{V}|}$ be the logit scores generated at the $t$-th step. Now, the watermarking operation can be described in the following steps:

1. First, the tokens in the vocabulary set $\mathcal{V}$ is divided into two disjoint subsets namely `red` and `green` list. This membership labeling is controlled by a context-dependent pseudo-random seed generated by hashing the set of tokens in the last $c$ time steps $\{s_{t-c}, \cdots, s_{t-1}\}$, where $c$ is the context width. In this paper, the context width is mainly set as $c = 1$, i.e., only the token in the previous time step $s_{t-1}$ is used for modulating the seed. Based on this division, the `green` list consists of $\gamma|\mathcal{V}|$ tokens where $\gamma \in (0, 1)$ represents the ratio of tokens in `green` list to `red` list.

2. For sampling $s_t$, Kirchenbauer et al. (2023a) discusses two major schemes to encode a watermark based on these *green* lists.: (a) "Hard" watermarking, and (b) "Soft" watermarking. Hard watermarking is designed to always sample a token from the `green` list and never generate any token from the `red` list. A major drawback is that for low entropy sequences where the next token is almost deterministic, hard watermarking may prevent the language model from producing them, resulting in degradation of the quality of watermarked text. To this end, Kirchenbauer et al. (2023a) proposes a "softer" version as their main watermarking strategy, in which for every token belonging to the `green` list, a scalar constant ($\alpha$) is added to the logit scores:

$$\tilde{\ell}_t[v] = \ell_t[v] + \alpha \, \mathbb{I}[v \in \texttt{green}] \tag{6}$$

where $\mathbb{I}$ indicates an indicator function. This helps in creating a bias towards sampling tokens from `green` lists more frequently as compared to `red` lists. After modification, the logit scores are generally passed through a softmax layer and the token $s_t$ is sampled from the modified distribution with any desired sampling scheme.

Given a text passage, to check for a watermark one does not need access to the source model as the random seed can be re-generated from the previous tokens. To identify whether a text sequence is watermarked or

not, one needs to simply recompute the context-dependent pseudo-random seeds and count the number of green tokens. Let $\mathbf{s} = \{s_1, s_2, \cdots, s_N\}$ represent a test sequence of length $N$. Then the number of green tokens in $\mathbf{s}$ is calculated as $|s_G| = \sum_{i=1}^{n} \mathbb{I}[s[i] \in \texttt{green}]$ where $\mathbb{I}$ represent the indicator function. The detection is then carried out by evaluating the null hypothesis which assumes that the sequence is generated by a natural writer with no knowledge of the $\texttt{green}$ or $\texttt{red}$ list. To test the null hypothesis, $z-$score is calculated as:

$$z = (|s_G| - \gamma N)/\sqrt{N\gamma(1-\gamma)} \tag{7}$$

where $\gamma$ represents the $\texttt{green}$ list ratio. A text passage is classified as "watermarked" if $z \geq \delta$, where $\delta$ is the classification threshold.

When the null hypothesis is true, i.e, the text sequence is generated by a natural writer, $|s_G|$ (number of $\texttt{green}$ tokens) follows a binomial distribution with a mean of $\frac{N}{2}$ and variance of $\frac{N}{4}$. This is because writings without knowledge of the watermarking scheme would has a $\gamma$ (e.g. 50%) probability of each token being sampled from a $\texttt{green}$ and $1-\gamma$ from a $\texttt{red}$ list. With an increase in the number of tokens, based on Central Limit Theorem, $s_G$ can be approximated with Gaussian distribution and thus can be approximated with $z$ score. Human-generated text without the knowledge of the green list rule is expected to have a lower number of $\texttt{green}$ tokens and thus a lower $z-$score as compared to machine-generated text. For shorter texts the Gaussian assumption does not hold anymore, however, Fernandez et al. (2023) has shown that detection is still feasible using an actual test of the true binomial distribution. Kirchenbauer et al. (2023a) and Fernandez et al. (2023) also both point out that in case a document consists of repetitive texts, the detection scores defined above might be erroneous. This is because the independence criterion necessary for calculating the $z$-scores does not hold in self-similar texts, with many repeating contexts, thereby artificially modifying the score. As a solution, Fernandez et al.; Kirchenbauer et al. propose scoring only those tokens during detection for which the watermark context plus the token itself ($\{s_{t-c}, \cdots, s_{t-1}, s_t\}$ have not been previously seen and Fernandez et al. (2023) show that this modification, in tandem with testing against a binomial distribution, leads to analytic false positive rates that closely match empirical estimates.

**Provable Robust Watermarking for AI-Generated Text (Zhao et al., 2023a)** Recent studies by (Sadasivan et al., 2023; Krishna et al., 2023) have questioned the robustness of watermarking-based detection models against modifications of the text. In light of that, Zhao et al. (2023a) proposed a robustified watermark-based framework named $\texttt{GPTWatermark}$. $\texttt{GPTWatermark}$ is similar to the watermarking scheme introduced in Kirchenbauer et al. (2023a), except that for every token they use a fixed split controlling the $\texttt{green}$ and $\texttt{red}$ list, i.e. context width of 0, whereas in Kirchenbauer et al. (2023a), the split is controlled by a pseudo-random seed based on the hash of the previous tokens.

A fixed split as in Zhao et al. (2023a) is optimal in terms of robustness to edits of the text. This robustness comes with a trade-off in detectability. As pointed out in Christ et al. (2023a), the length of the context (and entropy of the sequence) determine how detectable a watermark signal is, i.e how likely an attacker is to recover the watermark from observing watermarked text. Detectability also implies that impact on text generation quality or impact on users (who might observe that the model is less likely to use certain words) cannot be ruled out, but Zhao et al. (2023a) show that that these effects are likely small in practice.

As an improvement over Kirchenbauer et al. (2023a), Zhao et al. also provides a theoretical understanding of the robustness properties of the watermarking scheme. As a threat model, the paper considers adversaries with only black-box access to the language model. Given an adversary $\mathcal{A}$ and a watermarked sequence $\mathbf{s} = \{s_1, s_2, \cdots, s_n\}$, let the adversarially modified output be given as $\mathbf{s}_{\mathcal{A}}$. Zhao et al. assume that the edit distance between $\mathbf{s}$ and $\mathbf{s}_{\mathcal{A}}$ is upper bounded by some constant $\eta$ such that $\mathbf{ED}(\mathbf{s}, \mathbf{s}_{\mathcal{A}}) < \eta$. Where $\mathbf{ED}$ represents the edit distance and is defined as the number of operations required to transform $\mathbf{s}$ into $\mathbf{s}_{\mathcal{A}}$. Then the maximum text edit distance required to evade the $\texttt{GPTWatermark}$ can be given by:

$$\eta = \frac{\sqrt{n}(z_{\mathbf{s}} - \delta)}{1 + \frac{\gamma}{2}} \mathbb{I}\left[z_{\mathbf{s}} - \delta < \frac{\gamma\sqrt{n}}{1 + \frac{\gamma}{2}}\right]$$

where $z_{\mathbf{s}}$ represent the z-score corresponding to the watermarked sequence $\mathbf{s}$, $\gamma$ represents the green list ratio, and $\delta$ represents the classification threshold of the detector. Similarly, the maximum edit distance required

to evade the baseline watermarking scheme introduced in Kirchenbauer et al. (2023a) detector can be given by:

$$\eta = \frac{\sqrt{n}(z_{\mathbf{s}} - \delta)}{2 + \frac{\gamma}{2}} \mathbb{I}\left[z_{\mathbf{s}} - \delta < \frac{\gamma\sqrt{n}}{2 + \frac{\gamma}{2}}\right]$$

, when setting a context width of 1. The above expressions highlight that it takes twice as many edits to evade `GPTWatermark` as compared to Kirchenbauer et al. (2023a). Note that a number of attacks, such as paraphrasing and generative attacks, do not assume that an attacker would be restricted to make only $\eta$ edits to the text.

**On the Reliability of Watermarks for Large Language Models (Kirchenbauer et al., 2023b)** As a refinement of the baseline watermarking scheme introduced in Kirchenbauer et al. (2023a), in this paper, Kirchenbauer et al. (2023b) proposed a more robust and improved hashing scheme for generating watermarked text. Recall that in Kirchenbauer et al. (2023a), the hashing scheme focuses a context width of $c = 1$, i.e., for the $t$-th time step the random seed for splitting the vocabulary is generated by hashing the token at only the $t - 1$-th step. Let this hashing scheme be denoted as LeftHash. An alternative hashing scheme (denoted as SelfHash) also uses the token at the position $t$ itself in addition to the tokens on the left of $t$. Both scheme can then be used with arbitrary context width $c$. The analysis in Kirchenbauer et al. (2023b) suggests using smaller context widths $c$ provides more robustness to paraphrasing attacks, but related to findings in Zhao et al. (2023a) and Christ et al. (2023a), a larger context width leads to a less detectable watermark, which comes with a smaller impact on text quality and a robustness against attacks that attempt to reverse-engineer the watermark.

Formally, a hash function is defined as $f : \mathbb{N}^c \to \mathbb{N}$, which takes in a set of tokens $\{s_{t-c}, \cdots, s_{t-1}\}$ as input and outputs a pseudo-random number. Let $a \in \mathbb{N}$ be a secret salt value and $P : \mathbb{N} \to \mathbb{N}$ be an integer pseudo-random function. In this paper, Kirchenbauer et al. introduces the following hashing schemes:

1. Additive: This is an improved version of the hashing scheme introduced in Kirchenbauer et al. (2023a). Specifically, the context width used for hashing is increased. The hashing function is defined as $f(\mathbf{s}) = P(a \sum_{i=1}^{c} s_i)$. This form of hashing is not robust to attacks that can remove a token $s_i$ from the context.

2. Skip: This scheme of hashing only takes into account the leftmost token in the context: $f(\mathbf{s}) = P(as_c)$.

3. Min: For this scheme, as the name suggests, the hash function is defined as the minimum of the hash value generated using each token $s_i$ in the context: $f(\mathbf{s}) = \min_{i \in \{1, \cdots, c\}} P(as_i)$

Empirical study by Kirchenbauer et al. (2023b) shows that for larger context widths Min and Skip variants are much more robust to paraphrasing attacks as compared to the Additive hashing scheme. However, in terms of the diversity of the text generated, the Additive scheme ranks better as compared to other variants, line with Christ et al. (2023a).

Further, Kirchenbauer et al. (2023b) also discuss the question of detecting a watermarked text embedded inside a much larger non-watermarked passage. There, computing the $z$-score globally would not provide an accurate measure. In light of this, Kirchenbauer et al. (2023b) propose a windowed $z$-score evaluation for detecting watermarked sequences in long documents. Let $\mathbf{s}$ be a text passage of length $T$ consisting of watermarked tokens. The detection is carried out by first generating a binary vector $x \in \{0, 1\}^N$ indicating the membership label of each token. Let $p_k = \sum_{i=1}^{k} x_i$ be the sum of the binary hits and $\gamma$ be the green list fraction. Then the WinMax score is calculated as:

$$z_{\text{WinMax}} = \max_{i,j} \frac{(p_j - p_i) - \gamma(j - i)}{\sqrt{\gamma(1 - \gamma)(j - i)}} \quad \text{for} \quad i < j.$$

**Natural language watermarking via paraphraser-based lexical substitution (Qiang et al., 2023)**
Qiang et al. propose a watermarking framework by incorporating a paraphrase-based lexical substitution method. The core idea involves generating substitute candidates for each token in a sequence using a paraphraser and embedding them in a way to preserve the meaning. Let $s_i$ represent the $i$-th token in a sequence $\mathbf{s}$ to be encoded using the watermarking framework. Also, let $\mathbf{m} \in \{0,1\}^N$ be a fixed binary watermark sequence. The watermarking procedure involves the following steps: (1) First, using a pre-trained paraphraser, substitute candidates are generated for the token $s_i$. Let $y_i$ represent the most probable substitute token generated by the paraphraser. (2) Next, the algorithm checks the validity of $y_i$ as a substitute token for $s_i$. If $s_i == \text{Para}(y_i)$, where Para represents the paraphraser, then $y_i$ is considered a valid substitute token. (3) Next, the list $C = \{y_i, s_i\}$ is sorted in an ascending alphabetical order. Finally, the encoded token ($\tilde{s}_i$) is set using the watermark sequence $\mathbf{m}$ as: $\tilde{s}_i = C[\mathbf{m}[i]]$.

After the watermarked text is generated, one can check for the watermark in a given sequence by extracting the binary watermark sequence $\mathbf{m}$ using the technique discussed above. Text generated from sources other than the watermarked language model would result in a different binary sequence. The paper lacks a detailed analysis to understand the effect of paraphrasing attacks on this watermarking scheme.

### 4.1.3 Pseudo-Random Watermarks.

Pseudo-random watermarking schemes (Aaronson, 2023; Christ et al., 2023a; Kuditipudi et al., 2023a) operate by minimizing the distance between the watermarked and original distribution, with the aim of making the watermark both undetectable and unbiased in expectation. Next, we briefly summarize few frameworks leveraging pseudo-random watermarks for AI-GTD.

**Aaronson Watermark (Aaronson, 2023).** Aaronson proposes a watermarking scheme in which given the tokens generated till $t-1$-steps $\mathbf{s}_{1:t-1} = \{s_1, s_2, \cdots, s_{t-1}\}$, the token at the $t$-th step is sampled as $s_t = \arg\max_{v \in \mathcal{V}} r_v^{\frac{1}{p_t(v)}}$. Here $r_v \in [0,1] \; \forall \; v \in \mathcal{V}$ are secret real numbers generated using a pseudo-random function using a secret key $sk$, $r_v = f_{sk}(s_{t-c}, \cdots, s_{t-1}, v)$. Given a test sequence $\mathbf{s} = \{s_1, s_2, \cdots, s_N\}$, the presence of watermark is identified by thresholding on the detection score, which is calculated as:

$$d = \sum_{i=1}^{N} \ln \frac{1}{1 - r_i'}$$

where $r_i' = f_{sk}(s_{t-c}, \cdots, s_{t-1}, s_t)$. Fernandez et al. (2023) highlighted an interesting property of this watermarking scheme. In expectation over the randomness of the secret vector $\mathbf{r} = [r_1, \cdots, r_{|\mathcal{V}|}]$, the probability of $s_t$ being token $v$ is exactly $p_t(v)$, i.e, $\mathbb{E}[\mathbb{P}(s_t = v)] = p_t(v)$. It means, that in expectation over $\mathbf{r}$, this watermarking scheme does not bias the distribution.

**Undetectable Watermarks for Language Models (Christ et al., 2023a)** Prior watermarking approaches introduced in Kirchenbauer et al. (2023a;b); Zhao et al. (2023a) operate through altering the distribution of the model output thereby leading to degradation in the quality of text generated (Christ et al., 2023a). Instead, Christ et al. focus on developing a watermarking scheme such that a watermarked text is computationally indistinguishable from non-watermarked text, a property that further implies no degradation in text quality (otherwise the presence of the watermark would be detectable through observation of reduced quality). They formalize the cryptographic notion of a watermark where it is computationally intractable to distinguish a watermarked text from the original output, and then develop a watermark fulfilling these requirements. Effectively, the watermark of Christ et al. (2023b) is intractable to observe for any parties not in possession of the secret key used to encode it.

Given a vocabulary set $\mathcal{V}$, let $\bar{\mathcal{V}}$ represent the binarized version of the vocabulary set where each token $v \in \mathcal{V}$ is encoded as a binary vector in $\{0,1\}^{\log(|\mathcal{V}|)}$. Let $F_{sk}$ represent a pseudo-random function modulated by the secret key $sk \in \{0,1\}^\lambda$ where $\lambda$ represents the security parameter. Note, that the output of $F_{sk}$ can be interpreted as a real number in $[0,1]$. Now, let us consider the process of generating the $i$-th bit, given the previous bits $b_1, b_2, \cdots, b_{i-1}$ are already decided. Let $p_i(1)$ represent the probability of the $i$-th bit being 1 according to the original model. For the $i$-th bit, the watermarking model outputs $b_i = 1$ if

$F_{sk}(\{b_1, b_2, \cdots, b_{i-1}\}, i) \leq p_i(1)$ otherwise $b_i = 0$. Note, that the probability of the $i$-th bit being 1 is exactly $p_i(1)$ since the output of $F_{sk}$ is drawn uniformly from $[0, 1]$. Thus, the generated watermarked text follows the same distribution as the original text making it computationally indistinguishable.

For detecting the watermark in a given sequence, the secret key $sk$ is leveraged. Let $\mathbf{x} = \{x_1, \cdots, x_L\}$ represent a binary encoded test sequence. For each text bit $x_i$, the detector calculates a score using the current bit and secret key $sk$:

$$m(x_i, s_k) = \begin{cases} \ln(F_{sk}(r_i, i)) & \text{if } x_i = 1 \\ \ln\left(\frac{1}{1 - F_{sk}(r_i, i)}\right) & \text{if } x_i = 0 \end{cases}$$

where $r_i = \{x_1, \cdots, x_{i-1}\}$. Finally, the score is summed over all the text bits, $c(\mathbf{x}) = \sum_{i=1}^{L} m(x_i, s_k)$. Unlike watermarked text, where the value of $F_{sk}(r_i, i)$ is correlated with $x_i$, in non-watermarked text $F_{sk}(r_i, i)$ is independent of $x_i$. Thus the expected value of $c(\mathbf{x})$ would be larger in a watermarked text as compared to non-watermarked text.

Although undetectable, Christ et al. caution that watermarking schemes, in general, cannot be made *unremovable* and can always be evaded using strong generative attacks that modify the entire text. However, Christ et al. (2023a) do not provide any empirical analysis to understand the actual robustness of the proposed undetectable watermarking schemes against, e.g. partial paraphrasing attacks, leaving the empirical robustness to edits uncertain.

### 4.1.4 Retrieval Based Methods

**Paraphrasing Evades Detectors of AI-generated Text (Krishna et al., 2023).** In a recent study, Krishna et al. show that detection algorithms (Kirchenbauer et al., 2023a; Mitchell et al., 2023) despite impressive performance can be vulnerable against paraphrasing attacks. Hence, to protect against paraphrasing, in this paper Krishna et al. proposes a retrieval-based detection strategy. This defense approach requires a database that store all machine-generated text from a particular model, and as such is also a defense that can only be mounted by the model owner. Specifically, given an input prompt and corresponding output generated by the language model, the API needs to store both the prompt and output sequence in a database.

During inference, given a text sequence, similarity scores are calculated between the samples stored in the database and the given sequence. A high similarity indicates that the given sequence is generated by the same language model. Intuitively, human writings are more likely to achieve a lower similarity score as compared to machine-generated text (considering that the source and target models are the same). In general, this kind of retrieval-based detection framework is reliable and to some extent robust to possible shifts in the text caused by paraphrasing attacks (Kirchenbauer et al., 2023b), as text passages can be matched based on semantic features. This scheme can also be attacked and in a recent work, Sadasivan et al. (2023) show that, at a fixed detection length, five rounds of recursive paraphrasing can cause a 75% drop in the detection accuracy of retrieval-based methods. However, it remains unclear whether the failure of a semantic match means that the recursive paraphrasing has modified the text too strongly and removed its original meaning, or whether the text is functionally the same, and paraphrased in a way that the semantic matcher does not pick up on. More research is necessary in this direction to accurately characterize the robustness of retrieval-based detection.

Another question with retrieval-based detection are concerns about privacy. The use of this detection approach is not always realizable, as local limitations and regulations, such as GDPR, might limit the amount of data that can be stored by model companies (Krishna et al., 2023). Nevertheless, retrieval-based detection could be the most accurate detection approach in jurisdictions where it is feasible, and could also be employed during judicial proceedings, when parties may request access to the database from model companies.

## 4.2 Post-hoc Detectors

In contrast to prepared detection methods are approaches that can detect AI-generated text without preparation or even cooperation by the model owner. This tasks is generally believed to be harder than prepared detection, but much more broadly applicable. Especially in scenarios where a large number of freely available language models are used by independent or uncooperative actors, post-hoc detection is the only way to detect AI-generated text.

### 4.2.1 Zero-shot Detection

For zero-shot text detection, no access to machine-generated or human-written text samples is required. The core idea is that generic text sequences generated by a language model contain some form of detectable information that can be picked up and flagged by a detector. Detection under this category may be performed using a pre-trained language model which may not be similar to the source model, or through a fully separate statistical approach.

Given a text passage, common approaches for zero-shot text detection include: (1) statistical outlier detection based on entropy (Lavergne et al., 2008), perplexity (Beresneva, 2016; Tian, 2023), or n-gram frequencies (Badaskar et al., 2008), and (2) calculating average per-token log probability of the given sequence and then thresholding (Solaiman et al., 2019; Mitchell et al., 2023).

**GLTR: Statistical Detection and Visualization of Generated Text (Gehrmann et al., 2019).** To aid the detection of machine-generated text, Gehrmann et al. (2019) propose a statistical outlier-based detection framework named `GLTR`. The framework is based on the assumption that language models generate text by frequently sampling from highly probable words. For the purpose of automated text detection, three tests are undertaken. Specifically, given a sequence $\mathbf{s} = \{s_1, s_2 \cdots, s_N\}$, for every token they calculate: (1) the probability of generating the token, (2) the rank of the word in the generated distribution, and (3) the entropy of the generated distribution. A high score in the first two tests indicates that the generated token is sampled from the top of the distribution and a low score in the third test means given the previous context, the model is highly confident of the generated token. Formally, let the target language model be defined as $\mathcal{L}_\theta$, the detection scores are defined as:

$$d_{\text{prob}} = \frac{1}{N-1} \sum_{t=2}^{N} \mathbb{P}_{\mathcal{L}_\theta(\mathbf{h})}[s_t | \mathbf{s}_{1:t-1}] \tag{8}$$

$$d_{\text{rank}} = \frac{1}{N-1} \sum_{t=2}^{N} \text{rank}(\mathbb{P}_{\mathcal{L}_\theta(\mathbf{h})}[\cdot | \mathbf{s}_{1:t-1}], s_t) \tag{9}$$

$$d_{\text{entropy}} = \frac{-1}{N-1} \sum_{t=2}^{N} \sum_{i \in \mathcal{V}} \mathbb{P}_{\mathcal{L}_\theta(\mathbf{h})}[i | \mathbf{s}_{1:t-1}] \log \mathbb{P}_{\mathcal{L}_\theta(\mathbf{h})}[i | \mathbf{s}_{1:t-1}] \tag{10}$$

where $\text{rank}(p, b)$ refers to the operation of calculating the rank of token $b$ in any distribution $p$. The authors empirically show that, unlike language models, human writers tend to use low-probability (defined in Equation 8) and low-ranking words (defined in Equation 9) much more frequently. The framework relies on these differences for the detection of machine-generated text.

**DetectGPT (Mitchell et al., 2023)** Mitchell et al. (2023) argue that zero-shot detection frameworks (Solaiman et al., 2019) based on thresholding the log-probability of a text passage, fail to leverage information about the local structure of the probability function. To this end, they propose `DetectGPT`, a zero-shot algorithm for detecting machine-generated text. The working principle of the algorithm is based on the empirical observation that text generated from LLMs tends to occupy negative curvature regions of the model's log probability function. Specifically, given a sequence $\mathbf{s}$ and the source language model $\mathcal{L}_\theta$, `DetectGPT` first generates $k$ perturbations of the given sample (say $\tilde{\mathbf{s}}_i \; \forall i \in \{1, 2, \cdots, k\}$) using some perturbation function $q$. In the paper, perturbations are added using the T5 (Raffel et al., 2020) model. For detecting machine-generated text, the authors calculate a discrepancy metric defined as:

$$d(x, \mathcal{L}_\theta, q) = \frac{\log \mathcal{L}_\theta(\mathbf{s}) - \frac{1}{k} \sum_i \log \mathcal{L}_\theta(\tilde{\mathbf{s}}_i)}{\tilde{\sigma}^2} \tag{11}$$

where $\tilde{\sigma}^2$ represents the variance of the perturbed text. The paper shows that text generated from language models tends to have a higher discrepancy metric as compared to human-generated text, thereby supporting detection.

A major limitation of `DetectGPT` is that it assumes white-box access to model parameters, which is not always realizable in practice. Further compared to other zero-shot detectors based on statistical outlier detection methods, `DetectGPT` is far more computationally expensive. Also, Krishna et al. (2023) show that even a single paraphrasing pass with a different model can cause significant degradation in the detection accuracy of `DetectGPT`.

**DNA-GPT (Yang et al., 2023)** Zero-shot detection frameworks based on thresholding log-probability of a given sequence such as `DetectGPT` (Mitchell et al., 2023) suffer from a major limitation by assuming white-box access to the model parameters. This means that the detector has access to the generated probability distributions at each time step. However, this is a strong assumption as the token probability distributions are not always accessible such as in the GPT-3.5 model.

In response, Yang et al. (2023) propose a detection strategy for black-box cases where detectors are restricted to only API-level access. Let $\mathbf{s} = [s_1, \cdots, s_n]$ represent a given text sequence of length $n$. For detection, the sentence is first truncated into two parts: $\mathbf{s}^{(1)} = [s_1, \cdots, s_{\lceil \gamma n \rceil}]$ and $\mathbf{s}^{(2)} = [s_{\lceil \gamma n \rceil + 1}, \cdots, s_n]$, where $\gamma$ is a hyper-parameter representing the truncate rate. The core idea of this detection strategy is based on the assumption that the uniqueness of each language model is manifested in its tendency of generating comparable n-grams. Leveraging this assumption, the detection model keeps aside $\mathbf{s}^{(2)}$ and feeds the subsequence $\mathbf{s}^{(1)}$ as input prompt to the language model with the task of completing the sequence. Multiple outputs are generated based on the input $\mathbf{s}^{(1)}$. Let the set of output sequences be represented as $\Omega = \{\bar{\mathbf{s}}^1, \cdots, \bar{\mathbf{s}}^K\}$, where $K$ represents the number of output sequences generated. Under the black-box setting, the detection is carried out by comparing the n-gram similarity between the sequences in $\Omega$ and $\mathbf{s}^{(2)}$. Specifically, the detection score with black-box access is calculated as:

$$\text{BScore}(\mathbf{s}, \Omega) = \frac{1}{K} \sum_{k=1}^{K} \sum_{n=n_0}^{N} f(n) \frac{\text{n-grams}(\bar{\mathbf{s}}^k) \cap \text{n-grams}(\mathbf{s}^{(2)})}{|\bar{\mathbf{s}}^k||\text{n-grams}(\mathbf{s}^{(2)})|}$$

where $f(n)$ is a weight function for different n-grams. In the paper, the default parameters are set as $f(n) = \text{nlog(n)}$, $n_0 = 4$ and $N = 25$. Intuitively, a higher detection score indicates that the sequence is machine-generated.

In the white-box setting, where one has access to the generated token probabilities, Yang et al. (2023) leverages the generated probability distribution. Specifically, the detection score with white-box access is calculated as:

$$\text{WScore}(\mathbf{s}, \Omega) = \frac{1}{K} \sum_{k=1}^{K} \log \frac{p(\mathbf{s}^{(2)}|\mathbf{s}^{(1)})}{p(\bar{\mathbf{s}}^k|\mathbf{s}^{(1)})}$$

**Smaller Language Models are Better Black-box AI-GTD (Mireshghallah et al., 2023).** The authors in Mireshghallah et al. (2023) also target the problem of detecting AI-generated text in a black-box cross-model setting where the detector has no access to the source model parameters. For detection purposes, (Mireshghallah et al., 2023) leverage the local optimality test introduced in Mitchell et al. (2023). The assumption here is that machine-generated texts are likelier to be locally optimal as compared to human-written text. Given a sequence $\mathbf{s}$, first $k$ perturbations ($\tilde{\mathbf{s}}$) are generated using a perturbation model such as $T5$ (Raffel et al., 2020) model. The curvature is then calculated as: $d(\mathbf{s}) = \log \mathcal{L}_\phi(\mathbf{s}) - \frac{1}{k} \sum_i \log \mathcal{L}_\phi(\tilde{\mathbf{s}}_i)$. Note, unlike Mitchell et al. (2023) which assumes white-box access, $\mathcal{L}_\phi$ here refers to an unknown language model different from the source model. The paper evaluates a suite of 27 different detector models with varying numbers of parameters ranging from 70M to 6.7B. Based on empirical analysis, they observe two interesting trends:

- For cross-model detection, smaller models show stronger performance as compared to models with larger capacities. However, they also observe that overlap in architecture family and dataset between the generator and detector model leads to improvement in detection performance.

- Interestingly, empirical results highlight that partially trained models are better detectors than fully trained models. For this experiment, they save checkpoints at different steps during the training process. They find that the final checkpoint is consistently the worst one in terms of machine-generated text detection. The authors hypothesize that this might be due to overfitting associated with a longer training process.

### 4.2.2 Methods based on Training and Finetuning of Classifiers

Another line of work focuses on training a binary classifier using features extracted from a pre-trained language model for detecting machine-generated text (Fagni et al., 2021; Bakhtin et al., 2019; Jawahar et al., 2020; Chen et al., 2023). This approach has a longer history with hallmark studies concerning finetuning classifiers for detecting neural disinformation in Hovy (2016); Zellers et al. (2019). In 2019, (Solaiman et al., 2019) achieved a then state-of-art performance by finetuning RoBERTa models for the task of detecting webpages generated by GPT-2. Recently OpenAI's (OpenAI, 2023) work on machine-generated text detection by finetuning a GPT model has followed these development, although OpenAI's detector has now been taken offline, due to its high false-positive rate. Detectors under this category do not require access to model parameters and hence can operate under complete black-box settings. However, unlike the zero-shot setup, supervised training samples are required in the form of human and machine-generated text to train the detector.

Detectors under this category suffer from a few drawbacks: (1) Collecting sufficient data to train the classifier can be challenging, especially in diverse domains where the availability of training samples is a major bottleneck. (2) With recent advancements, text generated by language models has become increasingly similar to human-generated text, making detection harder (Zhao et al., 2023a). (3) False-positive rates for these detectors are hard to establish and depend crucially on the data distribution used to train the detector (Liang et al., 2023). Further, a recent study by Gambini et al. has shown that detection strategies designed for smaller models such as GPT-2 lose their efficacy when applied to larger models such as GPT-3. Wolff & Wolff (2020) has also questioned the robustness of detectors against adversarial attacks.

**GPT-Sentinel (Chen et al., 2023).** Targeting the problem of machine-generated text detection, Chen et al. (2023) propose two simple approaches: (1) Training a simple linear classifier on top of a pre-trained RoBERTa model (Liu et al., 2019), and (2) Finetuning a T5 (Raffel et al., 2020) model. For this purpose, Chen et al. (2023) curate a dataset namely `OpenGPTtext` by paraphrasing textual samples from the `OpenWebText` (Gokaslan et al., 2019) corpus using the GPT-3.5-turbo model. Specifically, `OpenGPTtext` consists of $29,395$ samples, where each textual sample corresponds to a human-written text from `OpenWebText` corpus. They demonstrate that both fine-tuning and linear-probing technique on pre-trained models result in enhanced performance in identifying machine-generated text in comparison to baseline techniques like those presented by Tian (2023), OpenAI (2023), and Solaiman et al. (2019). The fine-tuning approach exhibits better accuracy in text detection than linear-probing, i.e., simply training a linear classifier on top of the frozen pre-trained model.

**LLMDet: A Large Language Models Detection Tool (Wu et al., 2023a).** To detect machine-generated text, Wu et al. propose a text detection framework namely `LLMDet`. A previous study by Mitchell et al. (2023) has highlighted the usefulness of perplexity in detecting machine-generated text. However, calculating perplexity requires white-box access to the original model parameters which may be unrealizable in practice. To leverage the usefulness of perplexity without assuming white-box access, Wu et al. introduce the notion of calculating a proxy score for perplexity. The text detection framework consists of two phases: (1) the Dictionary phase, and (2) the Training phase. In the dictionary phase, n-grams and their corresponding probabilities are stored as keys and values respectively in a dictionary. First, the language model is prompted repeatedly to generate a collection of text samples. Next, n-gram word frequency statistics are calculated on the set of generated text. Finally, n-grams and their corresponding probabilities are stored as keys and

values respectively in a dictionary. The probability for a n-gram is calculated based on frequency statistics of text generated by the language model. In the training phase, information stored in the dictionary phase is used to calculate the proxy perplexity. Finally, these proxy perplexities are used to train a text classifier to distinguish between machine-generated text and human writing.

**How Close is ChatGPT to Human Experts (Guo et al., 2023)?** The contributions of Guo et al. are essentially two-fold: (1) First, they aim to shed light on how close are state-of-art LLMs to human writers. Specifically, they provide an understanding of the properties of text generated by ChatGPT and how it differs from human writings. (2) Second, to minimize risks related to AI-generated content, they propose different detection systems. To understand the differences in text generated by ChatGPT and human writers, they curate a Human ChatGPT Comparison Corpus. In the corpus, each instance is a question followed by answers from a human expert and a response generated using ChatGPT. Using this dataset, they conduct two specific tests. In the first test, referred to as the Turing test, given a question and corresponding response (can be from a human writer or ChatGPT), a human evaluator has to identify whether the response is machine-generated or not. In the second setup, referred to as the Helpfulness test, given a question and responses from a human writer and ChatGPT, the task is to evaluate which response among the two is more helpful. Based on this analysis, they observe: (1) Responses generated by ChatGPT are correctly identified by a human evaluator approximately 80% of the time. (2) ChatGPT-generated answers are in general more concrete and helpful than human writers specially for questions from the domain of finance and psychology. However, for questions related to medical domain ChatGPT generated answers are not much helpful and poorly crafted as compared to human writings.

Further, for the purpose of detecting AI-generated text, Guo et al. (2023) consider: (1) Training a logistic regression classifier on the features obtained from GLTR (Gehrmann et al., 2019) test, and (2) finetuning a strong pre-trained transformer, specifically RoBERTa (Liu et al., 2019). They found that the RoBERTa-based detector is more robust and also exhibits stronger detection performance as compared to GLTR.

**Ghostbuster: Detecting Text Ghostwritten by Large Language Models (Verma et al., 2023).** Several studies, as discussed in the previous subsection (Mitchell et al., 2023) have explored the detection of machine-generated text by analyzing token log-probabilities. However a common challenge in generating the token probabilities involves assuming whitebox access to the target model. In their recent work, Verma et al. (2023) propose Ghostbuster, a detection framework that does not require access to token probabilities from the target model. This enables detection of text generated from unknown models. To learn the token log probabilities, during training Ghostbuster passes each document through a series of less powerful language models. However, instead of performing classification directly based on generated log probabilities, the token probabilities are passed through a series of vector and scalar operations. These operations combine the token probabilities giving rise to additional synthetic features. To illustrate, consider $\mathbf{p}_1$ and $\mathbf{p}_2$ be the generated probability vectors for any two tokens. Some instances of vector operations include addition ($\mathbf{p}_1 + \mathbf{p}_2$), subtraction ($\mathbf{p}_1 - \mathbf{p}_2$), multiplication ($\mathbf{p}_1 \cdot \mathbf{p}_2$) and division ($\mathbf{p}_1/\mathbf{p}_2$). Scalar operations include computations like maximum ($\max \mathbf{p}$), minimum ($\min \mathbf{p}$), length ($|\mathbf{p}|$) and l-2 norm ($||\mathbf{p}||_2$). Apart from these synthetic features, Verma et al. proposed using a set of manually crafted features incorporating qualitative insights and heuristics observed in AI-generated text. Finally a logistic regression classifier is trained on top of these features to detect machine-generated text.

**RADAR: Robust AI-Text Detection via Adversarial Learning (Hu et al., 2023).** A significant challenge in identifying machine-generated text involves tackling paraphrasing attacks. Further, prior studies (Krishna et al., 2023; Sadasivan et al., 2023) have shown that (recursive) paraphrasing attacks can significantly reduce the detection performance. To mitigate this issue, Hu et al. (2023) proposed a novel detection framework called `RADAR`. This framework employs an adversarial learning approach to simultaneously train a detector and a paraphraser. At a higher level, detector's objective is to accurately differentiate between human-authored content and machine-generated text. In contrast, the paraphraser's training involves generation of plausible and realistic text with the aim of eluding detection by the detector. Let $\mathcal{L}_\theta, \mathcal{D}_\phi, \mathcal{G}_\sigma$ represent the target language model, the detector and the paraphraser parameterized by $\theta, \phi$, and $\sigma$ respectively. Note that the target language model $\mathcal{L}_\theta$ is frozen through-out and does not involve any train-

| Method | Dataset | Target Language Model | | | | |
|---|---|---|---|---|---|---|
| | | Dolly-V2-3B | GPT-J-6B | Dolly-V1-6B | LLaMA-7B | Vicuna-7B |
| log p | Xsum | 0.989 | 0.789 | 0.904 | 0.568 | 0.969 |
| | SQuAD | 0.983 | 0.754 | 0.909 | 0.599 | 0.910 |
| | WP | 0.990 | 0.892 | 0.890 | 0.865 | 0.976 |
| entropy | Xsum | 0.894 | 0.543 | 0.465 | 0.701 | 0.398 |
| | SQuAD | 0.502 | 0.566 | 0.501 | 0.613 | 0.394 |
| | WP | 0.214 | 0.312 | 0.209 | 0.404 | 0.189 |
| DetectGPT | Xsum | 0.985 | 0.768 | 0.820 | 0.678 | 0.883 |
| | SQuAD | 0.954 | 0.700 | 0.614 | 0.552 | 0.723 |
| | WP | 0.911 | 0.803 | 0.841 | 0.726 | 0.960 |
| OpenAI | Xsum | 0.905 | 0.917 | 0.945 | 0.832 | 0.790 |
| | SQuAD | 0.876 | 0.924 | 0.910 | 0.830 | 0.807 |
| | WP | 0.875 | 0.904 | 0.909 | 0.875 | 0.870 |
| RADAR | Xsum | 0.824 | 0.902 | 0.895 | 0.892 | 0.895 |
| | SQuAD | 0.692 | 0.734 | 0.820 | 0.715 | 0.895 |
| | WP | 0.697 | 0.735 | 0.808 | 0.734 | 0.814 |

Table 1: Evaluating popular language models using state-of-art Post-hoc detectors on Xsum, SQuAD, and WP dataset. The table is motivated from Hu et al. (2023). The values are obtained by reproducing the results in (Hu et al., 2023).

ing. The detector $\mathcal{D}_\phi$ and the paraphraser $\mathcal{G}_\sigma$ are initialized with pre-trained T5-large and RoBERTa-large models respectively. Let $\mathcal{H}$ represent a human-text corpus, generated by sampling $160K$ documents from WebText (Gokaslan et al., 2019). Let $\mathcal{M}$ be a corpus of AI-generated text generated using the target language model $\mathcal{L}_\theta$, by performing text completion using the first 30 tokens as prompt. The training procedure involves two components:

- **Training the paraphraser.** First, AI-generated text samples $\mathbf{s}_m \sim \mathcal{M}$ are fed to the paraphraser $\mathcal{G}_\sigma$ as input. Let $\mathbf{s}_p$ be the output of the paraphraser and $\mathcal{P}$ be the corpus of all paraphrased samples. The paraphraser parameters are updated using Proximal Policy Optimization (Schulman et al., 2017) with the reward feedback from the detector $\mathcal{D}_\phi$. The reward returned by $s_p$ is the output of $\mathcal{D}_\phi(s_p)$, i.e., the predicted likelihood of $s_p$ being human written.

- **Training the detector.** As in a typical GAN training, the detector is tasked with accurately distinguishing human-written samples from AI-generated and paraphrased counterpart. The detector is trained using logistic loss defined as: $L_{\mathcal{D}_\phi} = L_\mathcal{H} + \lambda L_\mathcal{M} + \lambda L_\mathcal{P}$, where $L_\mathcal{H} = -\mathbb{E}_{s_h \sim \mathcal{H}} \log(\mathcal{D}_\phi(s_h))$ represents the loss to improve the correctness of predicting $s_h \sim \mathcal{H}$ as human-written, $L_\mathcal{M} = -\mathbb{E}_{s_m \sim \mathcal{M}} \log(1 - \mathcal{D}_\phi(s_m))$ and $L_\mathcal{P} = -\mathbb{E}_{s_p \sim \mathcal{P}} \log(1 - \mathcal{D}_\phi(s_p))$ represents the loss to predict $s_m$ and $s_p$ from being predicted as human-text.

Empirical analysis highlights the strong text detection performance of `RADAR` on a suite of 8 target LLMs. Specifically on XSum dataset, when samples are paraphrased using a OpenAI GPT-3.5-Turbo API (which is different from the paraphraser using during training `RADAR`), `RADAR` improves detection performance by 16.6% and 59.5% as compared to a Roberta-based detector fine-tuned on WebText (Gokaslan et al., 2019) and `DetectGPT` (Mitchell et al., 2023).

## 5 Towards the Impossibilities of AI-generated Text Detection

### 5.1 Paraphrasing Based Attacks

#### 5.1.1 Red Teaming Language Model Detectors with Language Models (Shi et al., 2023)

Attacks introduced in Sadasivan et al. (2023); Krishna et al. (2023) evade detectors by paraphrasing the AI-generated text using a language model. However, for the proper functioning of the attack, they assume that the paraphrasing model is not protected by any detection mechanism. Shi et al. argues that this assumption is not always realizable in practice, as in the future all publicly available language models might have a detection framework in place as protection. To this end, Shi et al. (2023) proposes a more realistic attack mechanism for cases even when the paraphrasing model has a detection mechanism in place.

Given an input text prompt $\mathbf{h}$, let $\mathbf{s} = \{s_1, \cdots, s_N\}$ represent the output sequence of length $N$ generated by a language model $\mathcal{L}_\theta$. Let $\mathcal{G}_\phi$ represent another language model used for generating the attacks. Specifically, they consider two types of attacks:

- The first kind of attack involves perturbing the output sequence $\mathbf{s}$ to generate a modified output $\mathbf{s}'$. This is done by replacing certain tokens in $\mathbf{s}$ with substitute words maintaining the naturalness and original semantic meaning. For replacing a token $s_k$ in the original output $\mathbf{s}$, $\mathcal{G}_\phi$ is prompted to generate substitute candidates with the same meaning as $s_k$. Let $\tilde{\mathbf{s}}_k$ represent the set of substitute candidate tokens generated by $\mathcal{G}_\phi$. Given a substitution budget $\epsilon$, the minimization framework is defined as:

$$\mathbf{s}' = \arg\min_{\mathbf{s}'} \mathcal{D}(\mathbf{s}')$$
$$\text{s.t.} \quad s'_k \in \{s_k\} \cup \tilde{\mathbf{s}}_k,$$
$$\sum_{i=1}^{N} \mathbb{I}[s_i \neq s'_i] \leq \epsilon N$$

  where $\mathcal{D}$ represents the text detector being attacked and $\mathbb{I}$ represents an indicator function.

- The second kind of attack is based on perturbing the input prompt $\mathbf{h}$ to generate $\mathbf{h}'$, thereby leading to a modified output $\mathbf{s}'$. The main idea is to modify the input prompt to shift the distribution of the text generated by the language, thereby evading the detector. This is done by appending an additional learnable prompt $\mathbf{h}_p$ to the original input sequence $\mathbf{h}$, leading to a new prompt $\mathbf{h}' = [\mathbf{h}, \mathbf{h}_p]$. The additional prompt $\mathbf{h}_p$ is searched by querying the detector multiple times using $m$ input prompts $\{h_1, \cdots, h_m\}$. The objective function of the search is defined as:

$$\arg\min_{\mathbf{h}_p} \frac{1}{n} \sum_i \mathbb{I}[\mathcal{D}(\mathcal{L}_\theta([\mathbf{h}_i, \mathbf{h}_p])) \geq \delta]$$

  In simple words, the search function aims to find an additional prompt $\mathbf{h}_p$ such that the average detection rate is minimized for the new outputs generated. This form of attack is primarily targeted for detection frameworks based on fine-tuning or training a neural classifier.

Empirical analysis by Shi et al. (2023) shows that an attack based on modifying the output sequence causes an 88.6% degradation in AUROC for `DetectGPT`-based detector, a 41.8% improvement as compared to paraphrasing based attacks. Here the attack is based on GPT-2-XL model and XSum dataset.

#### 5.1.2 Paraphrasing Evades Detectors of AI-generated Text (Krishna et al., 2023).

The paper questions the robustness of current state-of-art text detection algorithms (Kirchenbauer et al., 2023a; Mitchell et al., 2023) against paraphrasing attacks curated to evade the detector. The authors argue that the current paraphrasing algorithms do not properly emulate real-world conditions and suffer from two

fundamental limitations: (1) They are trained on sentences, ignoring paragraph-level information. (2) They lack a knob to control output diversity. Hence, they might not be the ideal candidate to stress-test these text detection algorithms. Specifically as text is paraphrased more, it necessarily drifts away in meaning from the original text, making it less useful for an attacker. Previous paraphrasing attacks are trade-offs on this curve of textual similarity and paraphrasing strength. In this work, Krishna et al. (2023) use an explicit diversity parameter to control this trade-off. Improving on these limitations, Krishna et al. (2023) train an external paraphrase generator called DIPPER. The model is loaded from a T5-XXL (Raffel et al., 2020) checkpoint and is finetuned on the PAR-3 dataset (Thai et al., 2022) for paraphrase generation. Analysis by Krishna et al. (2023) shows that for the GPT2-XL model, strongest paraphrasing attacks using DIPPER can degrade the detection capabilities of watermarked-based (Kirchenbauer et al., 2023a) model and DetectGPT by 52.8% and 65.7% respectively.

### 5.1.3 Paraphrasing by Humans

In a related setup, likely to occur in practice, the "attacker" might just themselves be a human writer who is paraphrasing a watermarked text with the purpose of evading detection. Interestingly, analysis in Kirchenbauer et al. (2023b) shows that human writers are indeed strong paraphrasers and are more capable of fooling a detector as compared to machine-based paraphrasers, which in turn means that text paraphrased by human writers needs to be significantly longer, for the watermark of Kirchenbauer et al. (2023b) to be detected.

## 5.2 Generative Attacks

Generative attacks, or cipher attacks, were proposed in Goodside (2023) and are studied in Kirchenbauer et al. (2023a). These attacks exploit the capacity of large language models for in-context learning, prompting these models to modify their responses in a manner that is both foreseeable and readily reversible. The essence of these attacks lies in subtly manipulating the model's output in a controlled manner, such that the attacker can predict the outcome and also revert the model back to its original state or responses if needed. An interesting example of such carefully crafted generative attack is the emoji attacks shown by Goodside (2023). The attack strategy works by instructing the model to produce an emoji following each token it generates. The tricky part is that removing these emojis would randomize the red list for the subsequent tokens, thereby effectively evading detection by the watermark detector. Other examples of generative attacks studied in Kirchenbauer et al. (2023a) include prompting the model to replace all 'a' with 'e'. Even other examples are prompts that ask the language model to return outputs completely different from the inputs, for example in a language like Chinese or in a base64 encoding.

These types of attacks effectively work by prompting to model to use a cipher during generation, that completely changes the n-grams observed in the text, but is easily revertible by the attacker. Compared to other attacks, such as edits or paraphrases, which only modify some of the tokens in the response, and thereby dilute a signal present in the text, these attacks can change the generated text completely, thereby fully removing, e.g. the watermark of Kirchenbauer et al. (2023a). However, executing these attacks requires a language model capable enough to adhere to the instructed rule without overly compromising the quality of its output. Kirchenbauer et al. suggests that one viable defense strategy against these attacks could involve incorporating adverse examples of such instructions during the fine-tuning process, thereby training the model to decline such requests.

## 5.3 Copy-Paste Attacks

Introduced in Kirchenbauer et al. (2023b), this is a form of synthetic evaluation where AI-generated text passages are embedded within a longer human-written document. This results in sub-spans of text sequences having an abnormally high number of tokens from the green list. Kirchenbauer et al. modulates the attack strength using two parameters: (1) the number of AI-generated sequences inserted in the document, and (2) the fraction of the document that represents AI-generated text. The empirical analysis in Kirchenbauer et al. (2023b) shows that usual detection approaches struggle when faced with text that is diluted in this manner,

and need to be adapted to include windowing scheme to still detect the smaller sequences of AI-generated text embedded in the document.

### 5.4 Spoofing Attacks

When an adversarial human deliberately generates a text passage to be detected as AI-generated, it is referred to as a spoofing attack (Sadasivan et al., 2023). The goal of these attacks could be to trick a detector into falsely classifying derogatory human writings as AI-generated with the malicious intention of harming the reputation of an LLM developer. Sadasivan et al. also provides approaches for spoofing various detection mechanisms. Specifically for watermarked models (Kirchenbauer et al., 2023a), the attack involves computing a proxy of `green` list for some of the most commonly used words in the vocabulary. For a watermark with a context width of 1, the attack in Sadasivan et al. queries the watermarked model a large number of times (for OPT-1.3B model, $10^6$ queries are computed) and observes pair-wise token occurrences in the output to generate an independent estimate the `green` list for each context (or just each token for a context width of 1). Then, after learning the proxy for these `green` lists, an adversarial human can generate text to be misclassified as watermarked by inserting these token sequences.

However, the hardness of this attack is directly related to the context width of the watermark, and attacks on watermarks with longer context widths, e.g. a width of 4 as in Kirchenbauer et al. (2023b) have so far not been demonstrated. For the watermark of Christ et al. (2023a), this type of attack has been proven to be intractable.

For retrieval-based detection (Krishna et al., 2023), Sadasivan et al. proposes to give human-written passages to the paraphraser. Recall that in the detection scheme proposed by Krishna et al. (2023), the output of the paraphraser model is stored in a database. Thus, all the paraphrased human writings are also fed to the database. Now, during inference, a human-written passage would achieve a high similarity score with the paraphrased human writings in the database, thereby fooling the detector to misclassify the human-written text as AI-generated.

Sadasivan et al. (2023) also demonstrates successful spoofing attacks for zero-shot and trained detectors (like OpenAI's Roberta-base detector) where they first prepend the original human-written text with other texts that were poorly detected by the detectors. The detection performance drops drastically (drastic increase in FPR at a fixed TPR) by spoofing attack demonstrating the efficacy of the spoofing attack. Thus the interestingly designed spoofing attack is shown to be successful against a majority of the detection and watermarking schemes (Sadasivan et al., 2023) and needs careful consideration. Although successful, such attacks raise a natural question of the motivation and purpose of adversarial and spoofing humans and applications where such attacks are practical like social media spoofing, online chat spoofing, etc. and we discuss the scenarios in Section 6.6.

## 6   Open Questions

In this section, we emphasize several critical questions and directions pertaining to the research on AI-generated text detection which are still unsolved and need further discussion.

### 6.1   Regulating Large Language Model Watermarks.

There is an ongoing world-wide debate on whether some form of regulation should be imposed on large language models to prohibit their use for nefarious purposes such as spreading disinformation or impersonating individuals. A plausible solution could be to set up a regulatory body to watermark all publicly available language models and install a detection mechanism as protection. A potential variant of this approach is that all watermarked models share a universally standardized detection key. Otherwise, it is possible to circumvent the watermark detector by diluting the distribution of tokens via paraphrasing with a second model with a different watermark. For example, let $\mathbf{s}$ represent an output sequence generated by a language model $\mathcal{L}_\theta$ watermarked with a key $k$. Let $\mathcal{G}_\phi$ be a paraphrasing model, watermarked with a different detection key $k'$. Then although watermarked, paraphrasing $\mathbf{s}$ by $\mathcal{G}_\phi$ can still modify the distribution of tokens being

sampled from the `green` list, thereby evading detection of the first model, and only partially encoding the watermark of the paraphrasing model. On the other hand, regulation could also adopt a principle of "last model responsibility", where responsibility for AI-generated text falls onto the company whose watermark is detected, even if the text may be originally sourced from another model.

The potential of regulation is further highly related to the amount of democratization in the ecosystem of large language model. If only a few companies dominate and produce most of the AI-generated text, then coordination is easy. If text generation is split across many independent actors, then attackers might always prefer, or directly control, those models that are not watermarked.

From a broader perspective though, regulation of large companies may be a strong approach to reduce the overall impact of AI-generated text, even if it is only making detection easy in all non-adversarial scenarios, i.e. documentation of AI-generated text, accidental mis-attribution, etc. Smaller open-source models and smaller companies could be exempt until a significantly large fraction of AI-generated text is made by their models.

## 6.2 Accurate Evaluation of Detectors

In the past few years, the natural language processing community has witnessed a deluge of studies being published trying to solve the problem of AI-generated text detection. The majority of these studies, promote their proposed framework by showing an improved accuracy in distinguishing machine-generated text from human writings. In most of these works, generally, texts from datasets such as XSum (Narayan et al., 2018) and SQuAD (Rajpurkar et al., 2016) are used as a proxy to characterize the human distribution. However, there may be a huge diversity between the original and modeled human distribution. For example, from the context of text generation, the fluency of answers written by linguistic experts or native speakers will be very different from the ones authored by non-native speakers or writers. Further, the recent observation by Liang et al. (2023) that perplexity-based detectors are biased against non-native speakers highlights the urgency of an accurate characterization of human distribution to train these detectors. This naturally leads to a challenging question: *how can we accurately evaluate detectors on broader parts of the human language distribution?*

## 6.3 Conditions for AI-generated text Detection

The origin of debate regarding the possibilities and impossibilities of AI-generated text detection originated from Sadasivan et al. (2023). Specifically, Sadasivan et al. (2023) posit theoretical arguments stating that when there is a strong overlap between machine and human-generated text distributions, it is impossible to detect AI-generated text. However, recent work by Chakraborty et al. (2023) derive a precise sample complexity analysis and show that it is almost always possible to detect AI-generated text from information-theoretic principles if one can collect enough samples or increase the sequence length. Corroborating the analysis presented by Chakraborty et al. (2023), Kirchenbauer et al. (2023a) also recently show that increasing the number of tokens significantly plummets the success of a paraphrasing attack. This leads to a natural question: in which settings one can always obtain additional samples? In the case of detecting text generated by interactive Chatbots or Twitter bots, one can always prompt the model multiple times to obtain enough training samples for obtaining perfect detection accuracy. However, it may not always be the case. Consider an educational setup, where the task is to check for the originality of a student-authored essay. In the worst case, the distribution of the essays written by all students in a class may closely follow the distribution of machine-generated essays. Since, in this scenario, collecting additional samples is impossible, according to the analysis presented by Sadasivan et al. (2023), one can only do as good as a random detector. This leads to another open question: *can we improve detection performance without assuming the availability of additional samples?*

## 6.4 Improving the Fairness of Text Detectors.

Although there has been a deluge of studies in the last few years on improving machine-generated text detection, a recent study by Liang et al. (2023) highlights serious concerns regarding implicit biases present in

GPT-based detectors against non-native English writers. Specifically, Liang et al. (2023) observe perplexity-based detectors having a high misclassification rate for non-native authored TOEFL essays despite being nearly perfectly accurate for college essays authored by native speakers. Liang et al. hypothesize that a plausible explanation for this biased behavior could be because of low perplexity in text written by non-native English speakers due to a lack of linguistic variability and richness. Biases in text detectors can have serious repercussions in educational settings such as falsely accusing a marginalized group of plagiarism. Thus there is a dire need to focus on mitigating bias and improving fairness in text detectors based on perplexity scores or classifiers.

This problem is especially pronounced in black-box detection system based on finetuned classifiers, which depend strongly on the type of text they are trained on, but can also appear in zero-shot detectors if they are not carefully evaluated and tuned. On the flip-side, watermark detection is unaffected, as it is based on a null hypothesis that holds independently of the input text. Retrieval is likewise also unaffected, if the retrieval model is unbiased.

### 6.5 Improving the Robustness of Text Detectors.

Based on the comprehensive discussion of various attack frameworks, we make the following key observations: (1) It is clearly evident that recursive application of paraphrasing attacks can exert an adverse impact on the efficacy state-of-art text detectors (Sadasivan et al., 2023; Krishna et al., 2023). It is also important to note that while employing an iterative randomized paraphraser is guaranteed to eventually remove all traces of the original text including watermarks, it invariably introduces alterations in the original text's meaning. So, there is always a trade-off between the strength of the paraphrasing attack and the extent to which the text's meaning is shifted. (2) On the other hand, generative attacks exhibit a bijective nature and can even evade the detector without causing a substantial shift in text quality and meaning.

In light of these insights, how can we design a more robust detector or watermarking scheme against potential attacks still remains an open and critical challenge. Additionally, it is also imperative to acknowledge that the feasibility of the detection is inherently contingent on the objectives and constraints of the stakeholders involved. To illustrate, within an educational context, instructors might want to catch students who wrote essays using machine-generated text. In such scenarios, detection can be made easier if the instructor can beforehand generate a characterization of the distribution of machine-generated essays by querying the language model multiple times. There can be other scenarios, including: (1) Individuals aiming to accumulate evidence to ban LLM actors from social platforms such as Twitter, and (2) Those wanting to document language model usage to prevent dilution of the internet with machine-generated text. So, to summarize, the feasibility of detection is intrinsically linked to the specific requirements of the user such as the amount of text that needs to be detected, acceptable false-positive rates, and acceptable threshold of evidence.

### 6.6 Discussion on Different Detection Scenarios in Practice

The discourse on AI-generated text has been the focal point of extensive discussions recently, but the existing literature often lacks a comprehensive exploration of the diverse practical scenarios necessitating detection, their specific requirements, and the importance of accurate detection outcomes. Specifically, we note that it is imperative to discern the distinct demands depending upon the context, urgency, and gravity of detection in order to effectively understand both the potential and limitations of AI-generated text detection. In this section, we aim to shed light on a range of scenarios where the detection of AI-generated text is of utmost importance and connect it to the requirements of detection and available possible solutions in the literature.

1. **Fake News, Misinformation, and Social Media Content Moderation**
   - *Concern:* The potential for bias and misinformation spread through AI-generated content on social media platforms and reviews, typically via bots (De Angelis et al., 2023).
   - *Scenarios:*
     - *Social Media*: Need to detect AI-generated spam or malicious content for a safe online environment.

- – *User Reviews:* Detecting fake reviews generated by bots or AI to ensure the credibility of product and service reviews.
  - – *Information Verification:* The assessment of the authenticity of articles and posts is crucial in combating the dissemination of misinformation. Past studies (Pegoraro et al., 2023; Day, 2023; Gravel et al., 2023) have shown that language models can be employed to craft misleading fake articles there by necessitating robust verification techniques.
  - – *Deepfakes:* Detection of AI-generated text used in conjunction with deepfake videos or images is essential to counter disinformation campaigns. Deepfake technologies, when coupled with AI-generated text, can create compelling false narratives, posing a significant threat to public discourse and trust.
- *Note:* In the above scenarios, gathering multiple samples *at least* from the social bots is feasible. The samples may be collected from the same social bots or a group of identical bots. Based on the theoretical and empirical analysis in Chakraborty et al. (2023); Kirchenbauer et al. (2023b), the collected *i.i.d* samples can then be used further to enhance the reliability of AI-generated text detection methods.

2. **Academic Integrity**

- *Concern:* The potential misuse of AI in creating essays or assisting students dishonestly.
- *Scenarios:*
  - – Plagiarism Detection: Detecting AI-generated essays or reports submitted as original work by students in educational institutions is a persistent concern (Sullivan et al., 2023; Currie, 2023; Perkins, 2023).
  - – Cheating Prevention: Identifying automated tools used by students to cheat in online exams or assignments (Cotton et al., 2023).
- *Note:* In some specific scenarios such as in education systems, wrong detection can lead to false penalization of students, thereby causing drastic repercussions on their careers. Hence, for such scenarios, it becomes crucial to minimize false positives while detecting AI-generated content. However, we note that interestingly, in these scenarios, since the prompts are always available to the instructor, it is possible to learn the machine distribution by querying open-source models. Thus, detection becomes easier and boils down to identifying out-of-distribution samples in that context. Additionally, the instructor can ask for a minimum requirement of sentence length to enhance detection (Chakraborty et al., 2023). Thus, although the problem of detection is challenging, there are alternatives to deal with the same, which is an interesting direction for future research.

3. **Financial Services**

- *Concern:* The use of AI in generating phishing messages, especially when mimicking financial bodies (Bateman, 2020).
- *Scenarios:*
  - – *Fraud Detection:* The identification of AI-generated phishing emails or messages used in fraud attempts, especially when impersonating banks or financial institutions, is a critical application (Bateman, 2020).
  - – *Identity Verification:* Accurately determining whether text responses in identity verification processes are generated by humans or automated systems is vital for fraud prevention.
- *Note:* To minimize the threats, a possible solution can be: (1) watermark all publicly available language models, and (2) design email filtering schemes which can detect the embedded watermark in the phishing messages and flag them for further verification.

4. **Customer Support**

- *Concern:* Differentiating between AI and human-generated responses.
- *Scenarios:*

- *Chatbots*: In distinguishing between responses generated by AI chatbots and those from human customer support agents, transparency and quality assurance are key concerns (Jain et al., 2019). While it may be relatively easier to identify AI-generated text in an online interaction with AI agents, the potential presence of adversarial humans requires vigilance. The presence of adversarial humans in this scenario highlights the potential of a spoofing attack where adversarial humans can manipulate the chat to fool the detector into believing that it is a chatbot.
  - *Email Responses:* Identifying automated responses in email communication from businesses or customer service departments is essential to maintain customer trust and ensure efficient communication.
- *Note:* In the above scenarios, certain strategies, such as asking specific questions that AI may struggle to answer naturally, can aid detection.

5. **Intellectual Property**

- *Concern:* The unauthorized use of AI to reproduce copyrighted materials.
- *Scenarios:*
  - *Copyright Violation:* Detecting AI-generated text used in the unauthorized reproduction of copyrighted content, such as books or articles, is vital to protect intellectual property rights (Zhong et al., 2023). Automated tools can generate content that infringes on copyrights, necessitating robust detection mechanisms (Wu et al., 2023b).
- *Note:* A plausible solution would be to embed watermarks (Kirchenbauer et al., 2023a) into digital content, to make it easier to identify the source and provenance of the content.

## 6.7 Can We Leverage Our Knowledge of Input Prompts for Better Detection?

In this section, we look into a compelling question: *does foreknowledge of the input prompt (to be used by an adversary), simplify the detection task?* We contemplate that under certain circumstances, having advanced insight into the input prompt could prove advantageous. To illustrate, let's consider an educational context: during an exam, students are asked to compose an essay on "computers". Also, let us assume that the question is framed as: "Compose an essay on computers". In this setting, it would also be fair to assume, that students who are planning to resort to unfair means such as using a language model, would use the original question or a slightly paraphrased version as the input prompt. Then, for the purpose of detecting plagiarized essays, the instructor can beforehand compute a characterization of the distribution of machine-generated essays by querying the language model multiple times using the same question prompt: "Compose an essay on computers". Consequently, any student employing language models for writing the essay can be detected by calculating some form of similarity with the pre-established machine distribution. We believe studying how this approach can be expanded to other scenarios represents a promising avenue for future research.

## 6.8 Is Semantic Watermarking Possible?

All watermarking strategies discussed in this work depend on the form of a particular piece of text, not on its content. In this way, these watermarks are easy to deploy, as they are designed not to modify the content of the text, but are also fundamentally vulnerable to modifications of the text that change its form, but leave its semantic content unchanged, such as edits, paraphrases and generative attacks. This problem seems only solvable with watermarks that operate on not on the form of the text, but on higher-level semantics. How to design watermarks that operate on this level, while still including beneficial properties of non-semantic watermarks, such as undetectability, analytically defined p-values and low false positive rates, is so far an open question.

# 7 Conclusions

In this survey, we review works highlighting the possibilities and impossibilities of detecting machine-generated text. Specifically, we provide a precise categorization and in-depth analysis of various detection

frameworks and attacking schemes. To further benefit the ongoing research, we discuss challenging open research questions that still need addressing. We hope that our survey will be beneficial to the community in understanding the strengths and limitations of existing research related to AI-generated text detection.

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
