# OpenReview forum: "A Survey on the Possibilities & Impossibilities of AI-generated Text Detection"
_TMLR — Accepted by TMLR_

### Review · Reviewer_akKj · 2023-11-03

**Summary Of Contributions:**

This papers provides a comprehensive survey of the nascent problem of AI-generated text detection. The survey categorizes recent work on AI-generated text detection into methods showing possibilities of detection, such as watermarking and zero-shot techniques, and methods revealing impossibilities, like paraphrasing attacks. The paper provides a comprehensive taxonomy and analysis of detection schemes, attack strategies, theoretical insights, real-world applications, and open challenges. The paper is well written and is easy to read. I enjoyed reading this paper!

**Audience:**

Yes

**Broader Impact Concerns:**

None.

**Claims And Evidence:**

Yes

**Requested Changes:**

- Expand the comparison between detection schemes on criteria like usability, accuracy, and robustness. A table could help summarize.

- Provide more details and examples for generative and spoofing attacks.

- Aim for a balanced presentation of arguments around possibilities and impossibilities. Both have merits and limitations depending on the context.

- Include empirical evidence on real datasets for key detection schemes to substantiate claims of effectiveness.

- Shorten repetitive details when reviewing methods to improve flow and highlight novel contributions.

Overall, this is a comprehensive survey analyzing an important emerging field. Addressing the above suggestions would further improve quality and value to readers. I appreciate the authors' work and hope these comments are helpful.

**Strengths And Weaknesses:**

**Pros**:
- The paper provides a comprehensive taxonomy and analysis of recent work on AI-generated text detection. Categorizing methods into "towards possibilities" and "towards impossibilities" offers a useful framework.

- Thoroughly reviewing detection schemes like watermarking, zero-shot, and classifier-based methods gives a good overview of current techniques. Discussing attack schemes like paraphrasing, generative, copy-paste, and spoofing attacks highlights vulnerabilities.

- Connecting theoretical insights from recent studies provides useful context. The sample complexity analysis showing detection improves with more text is an important insight.

- The discussion of open questions identifies critical challenges like evaluating detectors, improving robustness, and leveraging prompt knowledge. This will help guide future work.

- Examples illustrating detection scenarios and requirements in areas like social media, academia, and customer support are insightful.

**Cons**:

- More comparison between the detection schemes on criteria like usability, accuracy, robustness would further highlight their tradeoffs.

- While paraphrasing attacks are analyzed in depth, discussions of generative and spoofing attacks could be expanded.

- The arguments around possibilities vs impossibilities could be presented in a more balanced manner, as both highlight important caveats.

- More empirical evidence demonstrating the effectiveness of detection schemes on real datasets would strengthen the claims.

---

> ### Author Response · Authors · 2023-12-18
> **Response to Reviewer akKj**
>
> **General Response** :  We sincerely appreciate the reviewer's detailed feedback on our paper, highlighting that our work produces a comprehensive taxonomy and analysis of recent work on AI-generated text detection. We want to thank the reviewer for acknowledging that our survey's theoretical and practical insights, including the open questions illustration of detection scenarios, are critical and can help in future research.
>
>
> **Response to Requested Changes**
> >Requested Changes 1 : Expand the comparison between detection schemes ... table could help summarize. Include empirical evidence on real datasets ... to substantiate claims of effectiveness.
>
> **Response to Requested Changes 1:** We agree that empirical evidence on real-world datasets for various SOTA detection schemes is critical to substantiate claims of effectiveness. Hence, we evaluate popular open-source LLMs using state-of-the-art Post-hoc detectors on Xsum, SQuAD and WP datasets in Table 1 (newly added), and the values are obtained by reproducing the results in [A].
>
> [A] Hu, Xiaomeng, Pin-Yu Chen, and Tsung-Yi Ho. "Radar: Robust ai-text detection via adversarial learning." arXiv preprint arXiv:2307.03838 (2023).
>
> >Provide more details and examples for generative and spoofing attacks.
>
> **Response to Requested Changes 2:** As requested, we have added additional descriptions regarding the generative and spoofing attacks in Section 5.2 and Section 5.4 with examples and applications in Detection Scenarios Section 6.6. For example - The generative attacks are made by instructing the model to generate emojis after each token or prompting the model to replace all ‘a’ with
> ‘e’, which evades the detection by the watermark detector by randomizing the red lists. Similarly, we highlight the effectiveness of various spoofing attacks on all detection schemes.
>
> >Requested Changes 3 : Aim for a balanced presentation .... possibilities and impossibilities.
>
> **Response to Requested Changes 3:** We have added more details and context in Section 3 in the *Hidden Possibilities of Text Detection* in favor of both possibilities and impossibilities, emphasizing that the claim of possibility or impossibility requires additional context about the problem.

---

> > ### Comment · Reviewer_akKj · 2023-12-18
> > **Thank You.**
> >
> > Thank You. These changes help improve the paper.

---

> > > ### Author Response · Authors · 2023-12-20
> > >
> > > Dear Reviewer,
> > >
> > > Thank you so much for your feedback.
> > >
> > > Regards,
> > > Authors

---

### Review · Reviewer_ZsKH · 2023-11-10

**Summary Of Contributions:**

A thorough literature review about recent work in AI-generated text detection.

**Audience:**

Yes

**Claims And Evidence:**

Yes

**Requested Changes:**

Would be nice to change to the title, but it's optional...

**Strengths And Weaknesses:**

Thanks to the authors for the hard work on this paper. I am not an expert in this sub-topic, so my comments are unfortunately limited.

Strengths
------------
- A thorough discussion of many recent papers
- Clear writing
- Good organization
- Nice outline of open questions

Weaknesses
----------------
- I recommend removing the word "Towards" from the title. I am not sure what it means to be going towards a survey. You are already there.

Typos
-------
- Page 4: "two major groups based on usability: preemptive" - do you mean "prepared"?
- Page 10: "with only black-box access to a language model Given an adversary" - missing period before "Given"?
- Page 10, bottom: "Kirchenbauer et al." has no year, while all other references do
- Page 13: "4.2.1 Zero-shotDetection" - missingspace
- Page 19: "an attack on based on modifying" - extra "on"
- Page 23: "6.6 Discuss About the Different Detection Scenarios" - should be "Discussion on Different Detection..."

---

> ### Author Response · Authors · 2023-12-10
> **Thank you for the constructive feedback**
>
> We sincerely appreciate the reviewer's thoughtful feedback on our paper. We would like to thank the reviewer for acknowleding the thorough discussion, clear writing and good organization. We have updated the title and have also incorporated the suggested writing changes in our updated manuscript.

---

### Review · Reviewer_ZHjB · 2023-12-03

**Summary Of Contributions:**

The paper is a review about the topic of detecting text generated by large language models. This is a very recent field with significant social impact. The paper organizes related works into two categories: possibility (how to detect) and impossibility (how to avoid detection). It starts with a summary of theoretical discussion on if such a detection is possible or not. There are arguments in both camps. Then the paper discussed algorithms. The detection algorithms split into watermarking techniques which need full access to the model internals, and blackbox detection that doesn't. The anti-detection algorithms use various paraphrasing techniques to remove the watermark or signature of distribution from the model generated text.

After introducing related works, the paper proposed a number of open questions for the research community to consider.

**Audience:**

Yes

**Broader Impact Concerns:**

There are no concerns about the broader impact of the work. This is an important topic to make sure LLM-era of AIs can be used in a positive way for the society.

**Claims And Evidence:**

Yes

**Requested Changes:**

- Add performance numbers to each algorithm, even just as reported, instead of saying "strong". Preferably in a table.

- Section 1.1. Be consistent about wording: "preemptive" vs "prepared"

- Section 1.1, "Towards the Impossibilities of AI-GTD", "three" major categories should be "four"?

- Figure 2. We should add world knowledge or context to the picture.

- Section 3, first paragraph, "text detectors [based on] [using] watermarking ..." repetitive.

- Section 4.1.2, second paragraph, "...first to propose [a a] watermark.." typo?

- Page 10 4th paragraph, "... blackbox access to the language model [G]iven an adversary..." should be lower case?

- Page 11, discussion about (Qiang et al 2023)'s method. The description of the algorithm is hard to understand. E.g. "s_i == Para(y_i)", does it mean one of the paraphrasing candidate of y_i matches, or top? And why do we need to sort list C in alphabetical order? Is there any pre/post processing to ensure fluency after token level substitution?

- Page 14 "Bot or Human..." This is more about bot detection than text detection. Reviewer feels that it's either should not be in this survey, or deserves its own category.

- Page 16. "GPT-Sentinel ..." last sentence, what is "linear-probing method"?

- Section 5.2 first paragraph, "... in a manner that is both [forseeable and readily reversible] ..." not clear what it means when first reading about it. Suggest some clarification right away.

- (minor suggestion) Page 20 last paragraph, "... Sadasivan et al. proposes to [input] human-written..." input -> give?

- Section 6.5 first paragraph, "(2) on other hand" -> "on the other hand"?

- Section 6.6 (1), Note section: "atleast" -> "at least"

- Section 6.6 (4), "potential presence of adversarial human" could you give an example?

- Section 6.7. Since the use of prompt/foreknowledge has been discussed multiple times in the paper, to this point the question feels quite repetitive to the Reviewer (although it is a valid research question). This could be another example of having better design of dimension in the taxonomy.

**Strengths And Weaknesses:**

The paper does a good survey of a new but important topic. Overall speaking the strength outweighs the weakness. The clarity in writing, both high-level taxonomy design and some details can be improved.

Strengths. The paper:

- Addresses a timely and important topic to the research community.
- Provides both theoretical and practical details of detection and anti-detection.
- Summarize and proposed a list of research questions

Weakness:

- Taxonomy design. The paper organizes all algorithms into detection/anti-detection which further split into different methodologies (paraphrase / watermark etc). However Reviewer can see that it's common for a paper/algorithm to be both in the detection/anti-detection category by focusing on a particular aspect, e.g. paraphrasing. When reading the paper, Reviewer got a feeling that techniques like paraphrasing get mentioned repetitively. Another example of "hidden" dimension is whitebox vs blackbox detection.

- Clear comparison of performance. The quality of detection is usually described as just "strong" without quantitative comparison to others. Reviewer understands that there are no consistent benchmarks and individual papers may not provide a readily available number. However it's still useful if the paper can give a tabulated overview with as detailed metrics as possible. Reviewer does notice that a better evaluation framework is one of the research questions proposed.

- Minor unclarity in writing. See the requested changes section for the list.

---

> ### Author Response · Authors · 2023-12-18
> **Response to Reviewer ZHjB**
>
> **General Response**:  We thank the reviewer for the thorough and detailed feedback on our paper, which helped improve the quality of our work. We also appreciate the reviewer's recognition of the significance and timeliness of our research in the rapidly evolving field of large language model text detection. The acknowledgment of our work as a comprehensive survey in this new and socially impactful area validates our efforts and highlights the importance of continued exploration and understanding in this domain, and we are thankful for the same.
>
> We respond to your suggestions in detail below and have incorporated the changes in our updated manuscript.
>
>
> **Response to Requested Changes**
>
> We have addressed all the minor corrections and typos suggested by the reviewer, including Section 1.1, Section 3, first paragraph, Section 4.1.2, second paragraph, Page 20 last paragraph, Section 6.5 first paragraph, Section 6.6 (1), Section 6.6 (4) etc.
>
> > Requested Changes1 : Add performance numbers to each algorithm, even just as reported, .... Preferably in a table.
>
> **Response to Requested Changes 1:** We have added the detection performance of several popular open-source LLMs using state-of-art Post-hoc detectors on Xsum, SQuAD and WP datasets in Table 1 (newly added) and the values are obtained by reproducing the results in [A].
>
> [A] Hu, Xiaomeng, Pin-Yu Chen, and Tsung-Yi Ho. "Radar: Robust ai-text detection via adversarial learning." arXiv preprint arXiv:2307.03838 (2023).
>
> > Requested Changes2 : Figure 2. We should add world knowledge or context to the picture.
>
> **Response to Requested Changes 2:** As correctly suggested by the reviewer, we update Text Detection Framework flowchart on Figure 2 by adding potential sources of world knowledge or context present in the detector for the detection performance like information from Social media, Literature, Scientific content etc.
>
> > Requested Changes3 : Page 14 "Bot or Human..." This is more about bot detection than text detection. Reviewer feels that it's either should not be in this survey, or deserves its own category.
>
> **Response to Requested Changes 3:** We agree with the reviewer and will remove the "Bot or Human? Detecting ChatGPT Imposters with A Single Question" from the Taxonomy as it needs a separate category about bot detection.
>
> > Reuested Changes4 : Page 16. "GPT-Sentinel ..." last sentence, what is the "linear-probing method"?
>
> **Response to Requested Changes 4:** Thank you for the comment. We have revised the sentence as "The fine-tuning approach exhibits better accuracy in text detection than linear-probing, i.e., simply training a linear classifier on top of the
> frozen pre-trained model."
>
> > Requested Changes 5 : Section 5.2 first paragraph, "... in a manner that is both [forseeable and readily reversible] ..." not clear what it means when first reading about it.
>
> **Response to Requested Changes 5:** We have added this explanation in Section 5.2- "The
> the essence of these attacks lies in subtly manipulating the model’s output in a controlled manner such that
> the attacker can predict the outcome and also revert the model back to its original state or responses if
> needed."

---

> > ### Comment · Reviewer_ZHjB · 2023-12-19
> > **Thanks for addressing the questions!**
> >
> > I have no more comments.

---

> ### Author Response · Authors · 2023-12-20
>
> Dear Reviewer,
>
> Thank you so much for your feedback.
>
> Regards,
> Authors

---

### Decision · Action_Editor_icf8 · 2023-12-21

**Recommendation:** Accept as is

**Comment:**

This is a strong survey paper about AI-generated text detection. It gives a thorough breakdown of attacks and defenses, discusses theory and practice, and covers both "possibilities and impossibilities". The reviewers gave some suggestions for how to improve it, but uniformly found it thorough, well-structured, and well-written.

**Audience:**

Yes, this is an in-depth, well-produced survey on a timely topic.

**Claims And Evidence:**

The reviewers found this to be a valuable survey.